# Impact of maternal HIV infection on the gut microbiome and metabolome of mothers and infants: the PRACHITi cohort in Pune, India

Jenna Mandell[1,9], Tian Wang[2,9], Jyoti S. Mathad[3], Mehr Shafiq[1], Shilpa Naik[4], Mallika Alexander [5], Vandana Kulkarni[5], Prasad Deshpande[5], Michael S. Humphrys[6], Bing Ma [6], Johanna B. Holm [6], Ramesh Bhosale[4], Khalil G. Ghanem[7], Aarti Kinikar[8], Jacques Ravel [6], Amita Gupta[7], Shuang Wang [2] & Rupak Shivakoti [1] ✉

Human immunodeficiency virus (HIV) affects millions of reproductive-age women globally, and during pregnancy is associated with adverse birth and infant health outcomes. Research on how maternal HIV shapes the gut microbiota, a potentially modifiable factor, during pregnancy, postpartum, and in infancy remains limited. The PRACHITi cohort study was conducted in India among 244 pregnant women with and without HIV, who were followed along with their children through 1 year postpartum. Our study focuses on secondary objectives of the PRACHITi study related to gut microbiota, with longitudinal samples being collected in the full cohort and more frequent sampling in a sub-study. Here, our findings reveal gut dysbiosis (based on 16S rRNA sequencing) and distinct plasma metabolomic profiles across pregnancy, postpartum, and their infants among women with HIV compared with seronegative women. We show that specific taxa and metabolites are differentially abundant by HIV status, some of which are linked to adverse outcomes, including preterm birth, low birth weight, and inflammation, conditions that are more common among populations with HIV. These results suggest potential biological pathways through which HIV affects maternal and infant health.

Human immunodeficiency virus (HIV) affects millions of women of childbearing age globally, with the highest burden in resource-limited settings where maternal morbidity and mortality are high[1,2]. With or without antiretroviral therapy (ART), HIV infection in pregnant women continues to be associated with increased adverse birth outcomes, including preterm birth, low birth weight, and small for gestational age[3–5]. Further, their children who are HIV-exposed uninfected (CHEU) also remain at increased risk for mortality, infectious morbidity,

[1]Department of Epidemiology, Columbia University Mailman School of Public Health, New York, NY, USA. [2]Department of Biostatistics, Columbia University Mailman School of Public Health, New York, NY, USA. [3]Department of Medicine, Weill Cornell Medical College, New York, NY, USA. [4]Department of Obstetrics and Gynecology, Byramjee Jeejeebhoy Government Medical College, Pune, India. [5]Byramjee-Jeejeebhoy Government Medical College-Johns Hopkins University Clinical Research Site, Pune, India. [6]Institute for Genome Sciences and Department of Microbiology and Immunology, University of Maryland School of Medicine, Baltimore, MD, USA. [7]Division of Infectious Diseases, Department of Medicine, Johns Hopkins University School of Medicine, Baltimore, MD, USA. [8]Department of Pediatrics, Byramjee Jeejeebhoy Government Medical College, Pune, India. [9]These authors contributed equally: Jenna Mandell, Tian Wang. ✉e-mail: rs3895@cumc.columbia.edu

impaired growth, and neurodevelopment compared to children who are HIV-unexposed uninfected (CHUU), i.e., children born to mothers without HIV[4,6].

The impact of HIV infection on the gut microbiome profile is an area of active investigation, as the gut microbiome can impact immunity, affect metabolic pathways (e.g., directly through production of metabolites or indirect impact on the pathways), and clinical outcomes[7,8], and is potentially modifiable. HIV infection, and specifically HIV infection of immune cells in the gastrointestinal tract, results in gut dysbiosis and epithelial barrier dysfunction, leading to gut and systemic inflammation, and metabolic dysfunction that are linked to adverse clinical outcomes[9]. Studies in non-pregnant adults with HIV have shown microbial shifts, including depletion of taxa *Akkermansia*, *Anaerovibrio*, *Bifidobacterium*, and *Clostridium*, which may contribute to chronic inflammation and metabolic dysfunction; notably, ART does not fully restore the gut dysbiosis[10]. Studies on the gut microbiota and metabolome are limited in pregnant and postpartum women with HIV (WHIV) and their CHEU.

A cross-sectional study of third-trimester pregnant women in Zimbabwe found reduced abundance of several taxa, including *Bifidobacterium* in WHIV[11], which was also noted in the microbiota of their CHEU in another study[12]. As the gut microbiota could change over the course of pregnancy and postpartum, with associated shifts in immunity and metabolism[13], longitudinal studies over pregnancy and postpartum, and in their CHEU are needed. Microbiota changes during this period have been linked to maternal and infant health outcomes, thus differences by HIV status could potentially explain the higher rates of adverse events observed in WHIV and their CHEU[14,15]. Understanding the longitudinal change in the gut microbiota over time could reveal critical timepoints during pregnancy and postpartum when interventions would be most effective in preventing adverse outcomes.

In the general population, the overall gut microbiota composition has been shown to be consistent over mid-to-late gestation, as assessed by α-diversity in the second and third trimester[16]. However, it is not known whether a chronic infection, such as HIV infection, results in distinct temporal dynamics of the microbiota during pregnancy (e.g., decreasing α-diversity as gestation progresses).

In this study, to address gaps in understanding how maternal HIV affects the gut microbiota in the perinatal period, we assessed the relationship between maternal HIV status and gut microbiota composition during the second and third trimesters, at 6 months postpartum, and their infants' gut microbiota at 6 months of age, using a longitudinal cohort of mothers with HIV on ART and without HIV, and their infants. We further studied the maternal plasma metabolomics during pregnancy in a subset to better understand differences in function and metabolic pathways in maternal HIV. To understand how the gut microbiota changes during pregnancy and by HIV status, we also nested a study in the cohort to conduct intensive bi-weekly (i.e., every 2 weeks) stool sampling during the second and third trimester in a subset of participants with and without HIV.

Our work shows pregnant and postpartum women with HIV, and their CHEU infants, exhibit gut dysbiosis and distinct metabolomic profiles compared with seronegative women, and their CHUU infants. Specific microbial taxa and metabolites are differentially abundant by HIV status, including those linked to adverse outcomes, such as preterm birth, low birth weight, and inflammation, conditions that are more frequently observed in populations with HIV. These findings identify potential biological pathways through which maternal HIV could affect both maternal and infant health.

## Results
### Sociodemographic characteristics and clinical features
Table 1 shows the sociodemographic and clinical characteristics by HIV status for the 207 pregnant women (75 WHIV (36%) and 132 seronegative for HIV (SN) (64%)) with available data at the third trimester. The median (interquartile range: IQR) age of these study participants was 23 (21–27) years. About 41% had anemia, 89% had an education of high school or less, 11% were past smokers, and 9% had gestational diabetes. There were significant differences for age by HIV status, with older age in WHIV ($p = 0.022$). There were more individuals with anemia among WHIV (45%) than among SN (37%) ($p = 0.097$), and more undernutrition (36% vs 28%, $p = 0.28$). The median (IQR) CD4 count for women with HIV was 439 (306.5–698) cells/mm³. The median (IQR) HIV viral load (VL) for those with detectable VL was 460 (103.75-1584.25) copies/mL. All WHIV were on ART, with 73% on TDF/3TC/EFV, 11% on AZT/3TC/NVP, and the rest were on another type of ART regimen. The median number of sexual partners in a participant's lifetime was 1 for WHIV and SN. WHIV had no partners newly diagnosed with HIV (Table 1), and a cross-tabulation of maternal HIV status at the third trimester by number of sexual partners in lifetime is shown in Supplementary Table 3. Of those with data, there were no participants who used drugs or had alcoholic drinks in the past 2 years (Table 1). Study population characteristics for mothers in the second trimester and their infants can be found in Supplementary Tables 1 and 2.

### Temporal Dynamics of the gut microbiota α-diversity by HIV status during pregnancy
We first assessed data from the sub-study with intensive frequent sampling during the second and third trimester to understand the temporal dynamics of the gut microbiota during pregnancy overall, and by HIV status. In this sub-study, 71 pregnant women had a total of 474 samples taken every two weeks between enrollment in second trimester and their last third trimester study visit with a median of 7 samples per participant. Gestational age at sampling ranged from 14 to 37 weeks.

We assessed whether microbiota α-diversity, Shannon index specifically, was similar between the second and third trimester using the 71 pregnant women with frequent sampling. Using a paired t-test comparing mean α-diversity from all the second trimester samples to that of the mean of all third trimester samples (Supplementary Table 4), we observed no significant differences across all 71 pregnant women or by HIV status.

Using the same frequent samples, we also used linear mixed models (LMM) with longitudinal α-diversity measures through second and third trimesters as the outcome and gestational age (in weeks) as the explanatory variable, and treated subject as a random effect to examine whether α-diversity changes over time in pregnancy within SN and WHIV, separately. The coefficient of gestational age within WHIV is 0.005 ($p = 0.30$) and within SN is <0.001 ($p = 0.93$). This suggests that α-diversity does not change over time during the second and third trimester within either group.

With the conclusion that microbiota α-diversity does not differ between second and third trimesters, we then tested whether α-diversity differs by HIV status using all longitudinal samples of the same pregnant woman ($n = 71$) from the second and third trimesters with LMMs, where we have repeated α-diversity measures as outcomes and HIV status as the predictor. We observed that WHIV had significantly lower α-diversity ($p = 0.015$) than SN women between the second and third trimester (Fig. 1). We observed similar results with other α-diversity indices (e.g., Chao1 ($p = 0.002$) and Fisher ($p < 0.001$)). In summary, we observe that WHIV had lower α-diversity than SN pregnant women, with a consistent relationship through the second and third trimester of gestation.

### Gut microbiota dysbiosis in WHIV during pregnancy
Next, we compared the microbiota profile during pregnancy utilizing data from the full cohort. 242 women out of 244 women in the full cohort had available samples at the second and/or third trimesters. In permutational multivariate analysis of variance (PERMANOVA)

## Table 1 | Sociodemographic and clinical features of study participants by HIV status at third trimester (*N* = 207)

| Characteristic | N (%)ᵃ | | | |
|---|---|---|---|---|
| | Overall (*N* = 207) | Women with HIV (*n* = 75 [36%]) | Women Seronegative for HIV (*n* = 132 [64%]) | *P*-valueᵉ |
| Age, median (IQR)ᵇ | 23 (21-27) | 25 (21-27.5) | 22 (20-25) | *0.022*ⁱ |
| Anemia | | | | 0.097 |
| Yes | 82 (41) | 34 (49) | 48 (37) | |
| No | 118 (59) | 35 (51) | 83 (63) | |
| Missingʰ | 7 | 6 | 1 | |
| Education | | | | 0.99 |
| None to High School | 183 (89) | 67 (89) | 116 (89) | |
| Post High School to Postgraduate | 24 (11) | 8 (11) | 16 (11) | |
| Smoking Statusᵈ | | | | 0.65 |
| Yes | 23 (11) | 7 (9) | 16 (12) | |
| No | 184 (89) | 68 (91) | 116 (88) | |
| Undernutrition (MUAC)ᶜ | | | | 0.28 |
| Yes (< 23 cm) | 64 (31) | 27 (36) | 37 (28) | |
| No (≥23 cm) | 143 (69) | 48 (64) | 95 (72) | |
| Gestational Diabetes | | | | 0.80 |
| Yes | 18 (9) | 7 (10) | 11 (8) | |
| No | 185 (91) | 66 (90) | 119 (92) | |
| Missing | 4 | 2 | 2 | |
| CD4 Count, median (IQR) | -- | 439 (306.5-698) | -- | NA |
| ART Regimen | | | | NA |
| AZT/3TC/ATV | -- | 1 (1) | 0 (0) | |
| AZT/3TC/NVP | -- | 8 (11) | 0 (0) | |
| TDF/3TC/EFV | -- | 55 (73) | 0 (0) | |
| Otherᶠ | -- | 11 (15) | 0 (0) | |
| Not Applicable (HIV-) | -- | 0 | 132 | |
| HIV Viral Load (VL) | | | | NA |
| Undetectable VL (≤ 40 copies/mL) | -- | 47 (63) | -- | |
| Detectable VL (> 40 copies/mL) | -- | 28 (37) | -- | |
| Number of sexual partners in lifetime, median (IQR)ᵍ | 1 (1-1) | 1 (1-1) | 1 (1-1) | 0.18 |
| Partners newly diagnosed with HIV | | | | NA |
| Yes | 0 (0) | 0 (0) | 0 (0) | |
| No | 194 (100) | 64 (100) | 130 (100) | |
| Missing | 13 | 11 | 2 | |
| Alcoholic drinks in the past 2 years | | | | NA |
| Yes | 0 (0) | 0 (0) | 0 (0) | |
| No | 195 (100) | 64 (100) | 131 (100) | |
| Missing | 12 | 11 | 1 | |

## Table 1 (continued) | Sociodemographic and clinical features of study participants by HIV status at third trimester (N = 207)

| Characteristic | N (%)ᵃ | | | |
|---|---|---|---|---|
| | Overall (*N* = 207) | Women with HIV (*n* = 75 [36%]) | Women Seronegative for HIV (*n* = 132 [64%]) | *P*-valueᵉ |
| Ever used drugs (Cocaine, Heroin, etc) | | | | NA |
| Yes | 0 (0) | 0 (0) | 0 (0) | |
| No | 195 (100) | 64 (100) | 131 (100) | |
| Missing | 12 | 11 | 1 | |

ᵃ*N* is number of individuals.
ᵇIQR stands for Interquartile Range.
ᶜUndernutrition is based on Mid-upper arm circumference (MUAC) measurements.
ᵈPast smoking status of women.
ᵉ*P*-values were calculated using a two-sided Fisher exact test for categorical variables and a two-sided Wilcoxon rank sum test for continuous variables to determine the difference in characteristics by HIV status but does not account for differences in the missing category.
ᶠOther ART regimens include: TDF/3TC/LPV/r, TDF/3TC/ATV/r, ABC/3TC/NVP, and TDF/3TC/RAL.
ᵍThis data was only queried for those co-enrolled in a sub-study (*N* = 45).
ʰMissing or NA are excluded from percent calculations.
ⁱ*P*-values < 0.05 are significant and italicized.

adjusting for age, mid-upper arm circumference (MUAC), and education, the overall gut microbiota profile, as assessed by β-diversity Bray Curtis distance, was different by HIV status in the second ($p = 0.05$) and third ($p = 0.004$) trimester (Supplementary Table 5).

We then assessed the differences in α-diversity as well as abundance of specific taxa during pregnancy by HIV status using longitudinal microbiota data available from the full cohort ($n = 242$). These analyses included all samples available during pregnancy, i.e., 242 women from the full cohort with a sample collected at the second and third trimester each, and additional samples collected from the overlapping 71 women in the sub-study with frequent sampling, a total of 747 stool samples. Similar to the results from the sub-study with 71 pregnant women, in a LMM with repeated α−diversity, WHIV had lower α-diversity (although not significant) with HIV status having a coefficient of −0.059 ($p = 0.12$) for Shannon index (Supplementary Table 6), with similar and significant results for Chao1 ($p = 0.005$) and Fisher ($p = 0.002$), adjusting for covariates (same as those for PERMANOVA).

To assess whether specific taxa at the genus level are different by HIV status using available repeated samples from second and third trimesters in the full cohort ($n = 242$), we used the Analysis of Compositions of Microbiomes with Bias Correction (ANCOM-BC2) method, where we included HIV status in the model, adjusted for age, MUAC, and education and treated subject as a random effect, to compare the bias-corrected log absolute abundance of taxa between the two groups. The results revealed 4 bacteria with differential abundance by HIV status at FDR < 0.05 level (false discovery rate using Benjamini-Hochberg (BH) adjustment) (Fig. 2 and Supplementary Table 7). We observed that WHIV had significantly higher abundance of *Fusobacterium, Megamonas*, and *Lachnoclostridium* compared to SN (Fig. 2 and Supplementary Table 7). Conversely, the abundance of SCFA-producing *Lachnospiraceae_NK4A136_group* was lower in WHIV, and lower levels of this taxon have been associated with inflammation and poor immune function[17] (Fig. 2 and Supplementary Table 7). ANCOM-BC2 analysis also identified structural zeros, defined as taxa that were differentially present/absent in WHIV compared to SN women. 16 taxa were differentially absent in WHIV but present in SN, while 12 taxa were differentially present in WHIV but absent in SN (Supplementary Table 8).

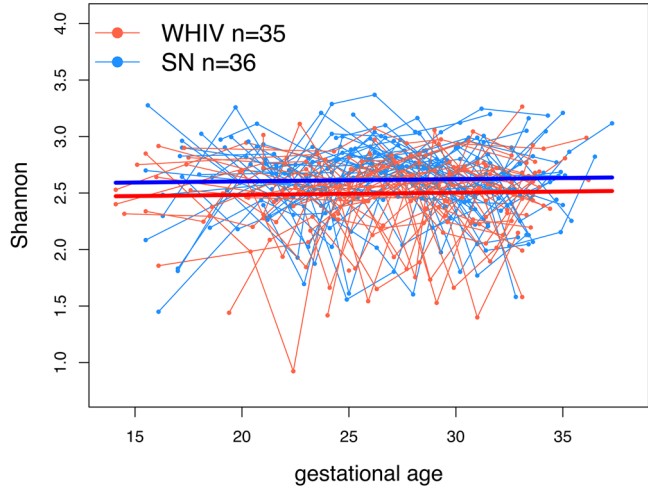

**coefficient of HIV positive status = -0.120 (p=0.015)**

**Fig. 1 | Gut microbial α-diversity among a subset of pregnant women with frequent sampling between second and third trimesters by HIV status (N = 71).** A spaghetti plot comparing α-diversity Shannon index over gestational age between pregnant WHIV (red) and SN (blue) with frequent sampling between the second and third trimester. Gestational age is on the x-axis and α-diversity Shannon index on the y-axis. The thick solid lines represent the model-estimated average α-diversity for WHIV and SN, while the thin lines show the α-diversity for each participant. The estimated coefficients and p-values (two-sided) are shown from a linear mixed effects model with α-diversity as the outcome, HIV status as the exposure variable, adjusting for gestational age, and treating subjects as random intercepts. The plot shows the estimated coefficients for HIV status and p-values in parentheses in the plot title. WHIV indicates women with HIV; SN indicates women seronegative for HIV.

Prior studies have noted increased levels of pro-inflammatory *Prevotella* in HIV[18], while *Bacteroides*, which includes many commensal gut bacteria important in SCFA production[19], may be lower. We specifically assessed these two bacteria to further understand their levels in WHIV during pregnancy and to test our hypothesis on whether the ratio of *Bacteroides* to *Prevotella* would be lower in WHIV. However, mean abundance of *Prevotella* (0.0002 vs. 0.0001 in WHIV and SN) and *Bacteroides* (0.08 vs. 0.05 in WHIV and SN) were both higher in WHIV compared to SN, although this association was not statistically significant in FDR-adjusted ANCOM-BC2 analyses on genus level taxa. We further conducted LMM with log ratio of *Bacteroides* to *Prevotella* in longitudinal samples as the outcome, HIV status as the exposure, and subject as the random intercept, and observed higher log ratio of *Bacteroides* to *Prevotella* in WHIV compared to SN ($p = 0.08$), which is not consistent with our hypothesis on *Bacteroides* to *Prevotella* ratio.

In an exploratory analysis, we also tested whether there was a difference in the third trimester microbiota of WHIV with detectable viral load (VL) as compared to WHIV with undetectable VL. We did not observe any significant differences in β-diversity ($p = 0.395$) or taxa abundance, but observed that the α−diversity Shannon index was higher among WHIV with detectable VL than undetectable VL ($p = 0.001$).

Further, in additional exploratory analyses, we compared the microbiota profile by HIV status in non-pregnant women and conducted another analysis comparing the microbiota between pregnant WHIV and non-pregnant WHIV to understand the impact of pregnancy among those with HIV (discussed further in Supplementary Results).

## WHIV have a distinct plasma metabolome profile in pregnancy
To better understand the functional changes by HIV status, we then compared the third trimester plasma metabolome in an analysis within a subset of 100 participants with plasma metabolome measures

(Fig. 3a). We identified 87 differentially abundant metabolites between WHIV and SN, with 40 metabolites at higher levels and 47 at lower levels in pregnant WHIV compared to SN. Some of the most significant metabolites included N2,N2-dimethylguanosine, cytosine, tetrahydrocortisol glucuronide, glucuronate, and methionine sulfone (Supplementary Table 9). These also included 5alpha-pregnan-3beta,20alpha-diol disulfate, 5alpha-pregnan-3beta-ol,20-one sulfate, and 5alpha-pregnan-3beta,20alpha-diol monosulfate, all of which are steroid sulfates[20]. Higher levels of N2,N2-dimethylguanosine, which was elevated in WHIV, show strong associations with preterm birth[21]. Further details on all significant metabolites in each group can be found in Supplementary Table 9. Among the 40 metabolites elevated in WHIV, the significantly enriched metabolites were from the following sub-pathways: fatty acid metabolism, N-acyl amino acids, pyrimidine metabolism, purine metabolism, corticosteroids, and food component/plant (Supplementary Table 9). Among the 47 metabolites lower in WHIV, these were from sub-pathways including sphingomyelin metabolism, progestin steroids, secondary bile acid metabolism, dihydrosphingomyelin metabolism, hexosylceramides, and androgenic steroids (Supplementary Table 9).

## Multi-omics among WHIV during pregnancy
Out of 100 pregnant women with plasma metabolome measures from the third trimester, 88 also have third trimester microbiota data. Using this subset of 88 pregnant women, we repeated the ANCOM-BC analysis and identified 14 differentially abundant genera with unadjusted p-values ≤ 0.1, and repeated a two-sample t-test and identified 87 significant metabolites. We then assessed correlations between abundance of the 14 microbes and metabolomic measures of the 87 significant metabolites (Fig. 3b and Supplementary Table 10). We found taxa Fusobacterium had the strongest correlations with several metabolites, including positive correlations with 2-hydroxyglutarate and phenol glucuronide, and inverse correlations with hydroxypalmitoyl sphingomyelin (d18:1/16:0(OH)), glycosyl ceramide (d18:1/20:0, d16:1/22:0), palmitoyl sphingomyelin (d18:1/16:0), and sphingomyelin (d18:2/23:1) (Fig. 3b and Supplementary Table 10). Taxa Ruminococcaceae_UCG-003 was positively correlated with metabolite 5alpha-pregnan-3beta,20alpha-diol monosulfate and pregnenolone sulfate (Fig. 3b and Supplementary Table 10). Additionally, the correlation of the 17 differentially present/absent structural zero genera from our ANCOM-BC analysis with the 87 significant metabolites is presented in Supplementary Table 11 and Supplementary Fig. 2.

## Gut dysbiosis persists for WHIV in the postpartum period
In our postpartum analysis of the full cohort, α−diversity Shannon index between WHIV ($N = 62$) and SN ($N = 120$) at 6 months postpartum was not significantly different ($p = 0.36$ using multivariable linear regression) (Supplementary Fig. 3). However, the overall gut microbiota assessed by β-diversity was significantly different between WHIV and SN ($p = 0.009$) (Supplementary Table 5). Across the three time periods by HIV status, α−diversity remained unchanged from second to third trimester but decreased slightly at postpartum (Supplementary Fig. 1).

There were no differentially abundant taxa at 6 months postpartum by HIV status with significant BH-adjusted p-value. However, the top two ranked taxa, *Lachnoclostridium* and *Megamonas*, are also top ranked and significant in the full cohort pregnancy analysis, pooling samples from second and third trimester samples, with higher abundance in WHIV. For the differentially present/absent taxa based on structural zeros, 34 bacteria were absent in postpartum WHIV but were present in SN women, including *Corynebacterium, Erysipelotrichaceae,* and *Granulicatella* (Table 2). Taxa *Peptostreptococcus* were differentially present in postpartum WHIV but absent in SN women (Table 2). In an additional analysis, we observed that the gut microbiota profile of postpartum WHIV more closely resembles that of

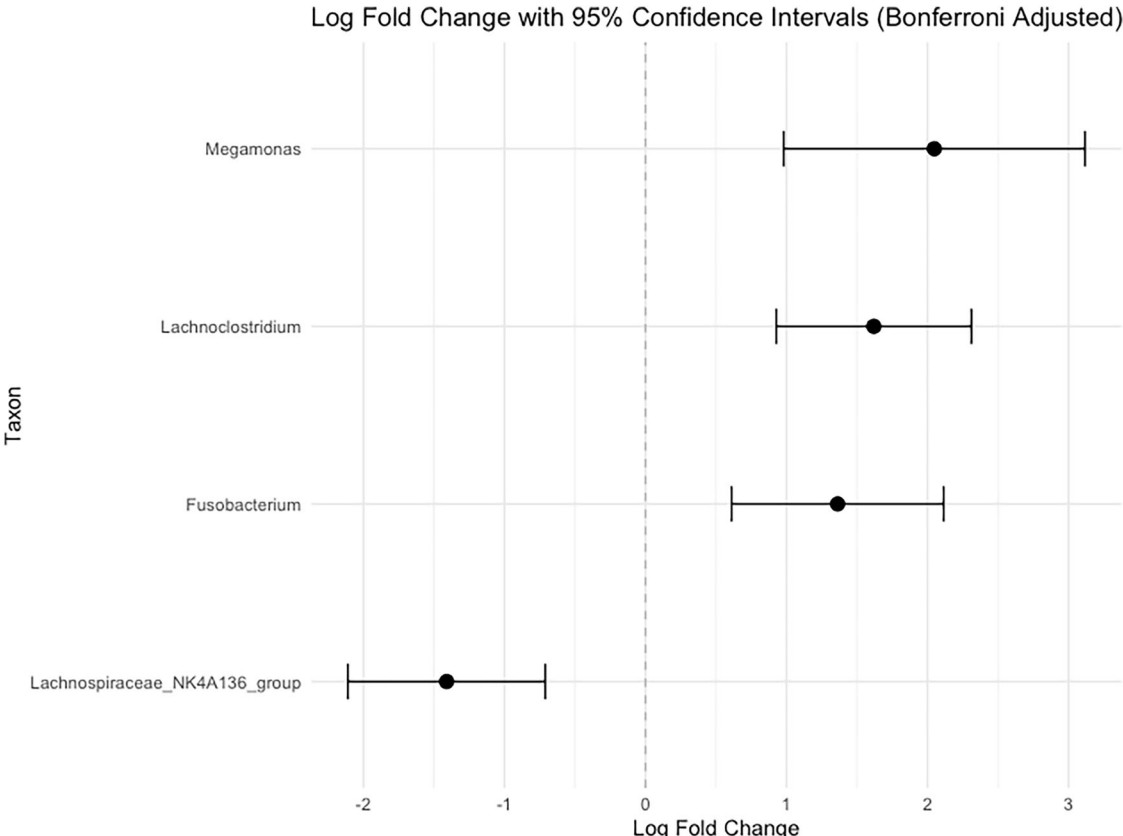

**Fig. 2 | Differences in the abundance of taxa at the genus level using all samples from the second and third trimester (including sub-study) by HIV status (N = 242).** The log fold-change with 95% confidence intervals (two-sided) plot represents the gut microbiota profiles of all samples from the second and third trimester by HIV status using the ANCOM-BC method with pseudo-count 0.1. It shows the Bonferroni adjusted log fold-change value (black dot) on the x-axis with the bars representing the lower and upper bound of the 95% confidence interval. This is shown for each statistically significant taxa (y-axis) from the Benjamini-Hochberg adjusted p-values (threshold of <0.05) in WHIV during the 2nd and 3rd trimester for Model 1, adjusted for age, mid-upper arm circumference (MUAC), and education, with subject as random effect. The log fold-change (LFC) represents the log change in corrected absolute abundance of bacteria in Model 1 (e.g., higher or lower) among pregnant WHIV compared to SN pregnant women. Error bars indicate 95% confidence intervals (CI) of the LFC estimates. LFC and confidence interval bars to the left of zero represent a lower abundance of bacteria in the WHIV group compared to the SN group, and LFC and confidence interval bars to the right of zero represent a higher abundance of bacteria in the WHIV group compared to the SN group.

non-pregnant WHIV, as no significant differences between groups were observed.

## Maternal HIV status impacts CHEU gut microbiota profile at 6 months old

Analysis of infants' gut microbiota at 6 months of age between CHEU (n = 63) and CHUU (n = 116) based on α− and β−diversity did not show significant differences (Supplementary Fig. 4 and Supplementary Table 5).

There were no differentially abundant taxa between CHEU and CHUU in the ANCOM-BC analysis with significant BH-adjusted p-values. However, 42 taxa were differentially absent in CHEU but were present in CHUU (Table 3). These include *Klebsiella* (also differentially absent in postpartum WHIV), *Ruminococcaceae_UCG-008*, and several from the *Prevotellaceae* family. Two taxa were differentially present in CHEU but were absent in CHUU, *Epulopiscium* and *Neisseria* (Table 3).

In our cohort, 86% of infants were being breastfed, and 34% exclusively breastfed at 6 months of age, with a slightly higher proportion of CHEUs (36%) exclusively breastfed at 6 months than CHUUs (33%). More information about taxa abundance related to exclusive breastfeeding can be found in the Supplementary Results.

## Discussion

Our study examined the impact of maternal HIV infection on the gut microbiota and plasma metabolome during pregnancy and

postpartum, and in their CHEUs. Our results from the sub-study with intensive sampling suggested that while WHIV consistently have lower α−diversity through gestation than SN women, the temporal dynamics of α-diversity remained consistent across gestation for both WHIV and SN. In the analysis from the full cohort, we found significant differences by HIV status, as assessed by β-diversity and taxa abundance analysis, during pregnancy and postpartum, with differences also reflected in their children. Key microbes such as *Lachnoclostridium, Fusobacterium, Megamonas*, and *Lachnospiraceae_NK4A136_group*, as well as key metabolites such as N2,N2-dimethylguanosine, 5alpha-pregnan-3beta,20alpha-diol sulfate, and its monosulfate variant were differentially abundant by maternal HIV status. In multi-omics analyses in a sub-sample during pregnancy, several correlations were observed between microbes and metabolites, with *Fusobacterium* positively correlated to 2-hydoxyglutarate and inversely correlated to several sphingomyelin metabolites. These microbes and metabolites could help explain the observed increased adverse outcomes in mother-infants with HIV as they have been linked to health outcomes such as inflammation, preterm birth, low birth weight, and neurodevelopment deficits. Future studies should confirm these findings and explore whether modulating these microbes or metabolites (e.g., through probiotics or prebiotics) could improve health outcomes of mothers and infants affected by HIV.

Using intensive stool sampling every 2 weeks from the second trimester through 37 weeks of gestation, our results showed

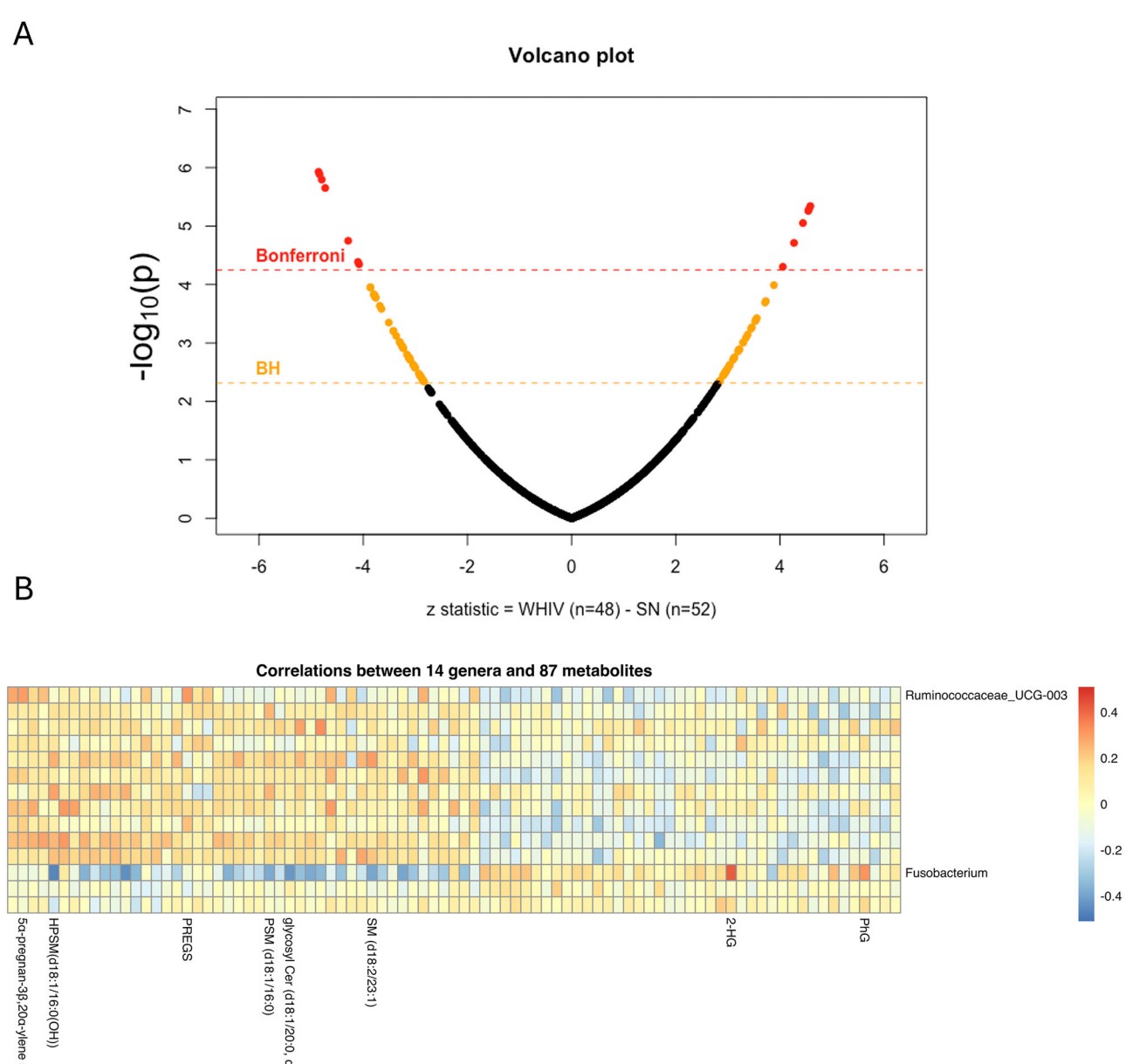

**Fig. 3 | Metabolomics (N = 100) and multi-omics (N = 88) analyses at the third trimester by HIV status. A** The volcano plot shows the differences in metabolite abundance between 100 WHIV and SN at the third trimester based on our metabolomics analysis using logistic regression with Model 1, adjusted for age, mid-upper arm circumference (MUAC), and education, after Benjamini-Hochberg (BH) adjustment, and points above this line are considered statistically significant. Bonferroni line (dashed red line) indicates the adjusted *p*-value (two-sided) threshold of 0.05 for significance. BH line (dashed orange line) indicates significance threshold adjusted to control the False Discovery Rate and account for false positives. The volcano plot is a visual representation of the most statistically significant metabolites. The *x*-axis represents the test statistic derived from comparing the abundance of metabolites between women with HIV (WHIV) and women seronegative (SN) for HIV, with the sign of the test statistic indicating the direction of the difference. The *y*-axis represents the negative logarithm (base 10) of the *p*-value comparing the abundances between the two groups. Red dots are metabolites significant after Bonferroni adjustment. Orange and red dots are metabolites significant after BH adjustment. Black dots are metabolites not significant after

either adjustment. **B** The heat map shows the correlation (two-sided) between 14 genera with unadjusted p-values ≤0.1 from our Analysis of Compositions of Microbiomes with Bias Correction (ANCOM-BC) analysis and 87 significant metabolites from our metabolomics analysis in a multi-omics analysis of 88 third trimester samples paired with metabolomics and microbiota data. Those in blue have a high negative correlation, and those in red have a high positive correlation. Metabolite and microbe associations with high negative or positive correlations are labeled. Due to the large number of metabolites and microbes, we only label select bacteria and metabolites with high positive or negative correlations. The specific correlations with an absolute correlation ≥ ±0.3 between microbes and metabolites are shown in Supplementary Table 10. Abbreviations: 5α-pregnan-3β,20α-ylene sulfate for 5alpha-pregnan-3beta,20alpha-diol monosulfate, HPSM(d18:1/16:0(OH)) for hydroxypalmitoyl sphingomyelin (d18:1/16:0(OH)), PREGS for pregnenolone sulfate, PSM (d18:1/16:0) for palmitoyl sphingomyelin (d18:1/16:0), glycosyl Cer (d18:1/20:0, d16:1/22:0) for glycosyl ceramide (d18:1/20:0, d16:1/22:0), SM (d18:2/23:1) for sphingomyelin (d18:2/23:1), 2-HG for 2-hydroxyglurate, and PhG for phenol glucuronide.

**Table 2 | Differentially present or absent taxa identified by ANCOM-BC[a] at 6 months postpartum by HIV status (N = 182)**

| Taxon | Structural Zero (WHIV)[b] | Structural Zero (SN)[c] |
|---|---|---|
| Actinobacillus | Yes | No |
| Anaerovibrio | Yes | No |
| Atopobium | Yes | No |
| Butyrivibrio | Yes | No |
| Candidatus_Soleaferrea | Yes | No |
| Coriobacteriaceae_UCG-003 | Yes | No |
| Corynebacterium | Yes | No |
| Elusimicrobium | Yes | No |
| Enterorhabdus | Yes | No |
| Erysipelotrichaceae_UCG-004 | Yes | No |
| F0332 | Yes | No |
| Gardnerella | Yes | No |
| GCA-900066225 | Yes | No |
| Gemella | Yes | No |
| Gordonia | Yes | No |
| Gordonibacter | Yes | No |
| Granulicatella | Yes | No |
| Holdemania | Yes | No |
| Hungatella | Yes | No |
| Klebsiella | Yes | No |
| Lachnospiraceae_NK4B4_group | Yes | No |
| Leptotrichia | Yes | No |
| Mogibacterium | Yes | No |
| Murdochiella | Yes | No |
| Neisseria | Yes | No |
| Oribacterium | Yes | No |
| Peptococcus | Yes | No |
| Prevotellaceae_UCG-001 | Yes | No |
| Ruminococcaceae_UCG-004 | Yes | No |
| Ruminococcaceae_UCG-009 | Yes | No |
| Sellimonas | Yes | No |
| Sneathia | Yes | No |
| Treponema_2 | Yes | No |
| Varibaculum | Yes | No |
| Peptostreptococcus | No | Yes |

[a]ANCOM-BC stands for Analysis of Compositions of Microbiomes with Bias Correction method and we used model 1 which adjusted for age, middle upper arm circumference (MUAC), and education.

[b]WHIV stands for women who have HIV. 'Yes' for structural zero for WHIV and 'No' in SN means the bacteria is differentially absent (i.e., absent in WHIV but present in SN).

[c]SN stands for seronegative HIV. 'No' for structural zero for WHIV and 'Yes' in SN means the bacteria is differentially present (i.e. present in WHIV but absent in SN).

consistently lower α−diversity in WHIV. Lower α-diversity was also observed in analysis of the full cohort, consistent with earlier findings in both pregnant and non-pregnant populations, and likely reflecting dysbiosis due to HIV infection[11,22]. The temporal dynamics of gut microbiota α-diversity, however, were consistent through gestation and did not differ by HIV status. Similar results were observed in populations without HIV, where the gut microbiota α-diversity did not change over gestation[16]. Although some studies have noted decreasing α-diversity as pregnancy progresses, there are important differences in the time-point of comparison data (i.e., notable differences between the first and third trimester, while ours compared the second and third trimester)[13,23]. While a disease or infection could modify the temporal dynamics of the microbiota profile over

gestation, our results suggest that WHIV on ART have a similar temporal profile to SN women.

In our full cohort analysis of the microbiota profile of pregnant WHIV and SN in both the second and third trimester, we noted gut dysbiosis at the taxa level by maternal HIV status. WHIV showed higher abundance of genera *Fusobacterium*, *Megamonas*, and *Lachnoclostridium*, and lower abundance of *Lachnospiraceae_NK4A136_group*. Aligned with these findings, higher levels of taxa from family *Veillonellaceae*, which *Megamonas* belongs to, during pregnancy have been associated with low birth weight in infants[11]. Additionally, an overgrowth of *Megamonas* could be involved in the development of gestational anemia[24]. Further, lower levels of SCFA-producing *Lachnospiraceae*, among those with disease conditions like HIV, are linked to poor gut health and immune function, inflammation, and disease progression[17]. Most research has linked *Fusobacterium* during pregnancy in the oral, vaginal, and placental microbiome with preterm birth[25]; however, a small number of studies in populations without HIV have also found higher abundance of gut *Fusobacterium* in women may potentially be linked to preterm birth[26,27]. Separately, in relation to HIV, enrichment of *Fusobacterium* has been associated with reduced immune recovery and persistent immune dysfunction, especially after ART regimens[28]. Our finding adds to the existing literature by showing higher abundance of *Fusobacterium* in pregnant WHIV.

In the third trimester metabolomic analysis, we observed differential levels of multiple metabolites, including increased levels of metabolites involved in fatty acid metabolism and purine metabolism pathways, and decreased levels of metabolites in progestin steroids and secondary bile acids pathways. We observed elevated levels of N2,N2-dimethylguanosine, a metabolite involved in purine metabolism, in WHIV. This metabolite is strongly associated with preterm birth and may help explain higher preterm delivery rates in this group[21]. Additionally, progestin steroid compounds, including 5alpha-pregnan-3beta,20alpha-diol sulfate and 5alpha-pregnan-3beta,20alpha-diol monosulfate, were lower in WHIV, consistent with previous findings among those on certain ART regimens[21].

In our multi-omics analysis, there were several notable correlations between specific microbes and metabolites that were significantly different between WHIV and SN. The strongest positive correlation was observed between *Fusobacterium* and 2-hydroxyglutarate, with both having higher levels than SN. HIV infection is known to affect glutamate and glutamine metabolism, which subsequently impacts immunity and neurotoxicity[29,30]. Increase in *Fusobacterium* and 2-hydroxyglutarate within WHIV might reflect this altered metabolism since *Fusobacterium* is known to ferment glutamate through the hydroxyglutarate pathway[31]. Further research is needed to better understand these relationships and their impact in pregnancy. The strongest negative correlations were observed between *Fusobacterium* and several sphingomyelin and related metabolites. *Fusobacterium* contains sphingolipids[32], thus higher levels of *Fusobacterium* could mean increased use of sphingomyelin by the bacteria and thus lower circulating levels of sphingomyelin. Alternatively, these and other relationships (e.g., positive correlation between *Ruminococcaceae_UCG-003* and 5alpha-pregnan-3beta,20alpha-diol monosulfate) could reflect indirect relationships between microbes and metabolites due to a shared pathway (e.g., inflammation[28,33,34]) impacted by HIV.

In the postpartum period, we observed significant differences in β-diversity by HIV status, consistent with prior studies at delivery and 62 weeks postpartum[15]. Although not statistically significant, postpartum WHIV had a higher abundance of *Megamonas* and *Lachnoclostridium*, bacteria that were significantly higher in WHIV during pregnancy. We observed several taxa to be differentially absent in postpartum WHIV, while genera *Peptostreptococcus* was differentially present among postpartum WHIV. The results related to *Peptostreptococcus* have been observed previously in WHIV and CHEU

**Table 3 | Differentially present or absent taxa identified by ANCOM-BC[a] at 6 months of infant age by maternal HIV exposure status (N = 177)**

| Taxon | Structural Zero (CHEU)[b] | Structural Zero (CHUU)[c] |
|---|---|---|
| Actinobacillus | Yes | No |
| Aggregatibacter | Yes | No |
| Aliihoeflea | Yes | No |
| Asteroleplasma | Yes | No |
| Barnesiella | Yes | No |
| Cutibacterium | Yes | No |
| Dysgonomonas | Yes | No |
| Enhydrobacter | Yes | No |
| Enterorhabdus | Yes | No |
| Erysipelotrichaceae_UCG-004 | Yes | No |
| Family_XIII_AD3011_group | Yes | No |
| Fournierella | Yes | No |
| Gordonia | Yes | No |
| Gordonibacter | Yes | No |
| Howardella | Yes | No |
| Klebsiella | Yes | No |
| Lachnospiraceae_UCG-010 | Yes | No |
| Marvinbryantia | Yes | No |
| Methanobrevibacter | Yes | No |
| Methanosphaera | Yes | No |
| Mogibacterium | Yes | No |
| Morganella | Yes | No |
| Murdochiella | Yes | No |
| Negativicoccus | Yes | No |
| Oribacterium | Yes | No |
| Oxalobacter | Yes | No |
| Paeniclostridium | Yes | No |
| Paraprevotella | Yes | No |
| Parvimonas | Yes | No |
| Peptostreptococcus | Yes | No |
| Prevotella_6 | Yes | No |
| Prevotellaceae_NK3B31_group | Yes | No |
| Prevotellaceae_UCG-001 | Yes | No |
| Pseudomonas | Yes | No |
| Rikenellaceae_RC9_gut_group | Yes | No |
| Ruminococcaceae_UCG-008 | Yes | No |
| Sellimonas | Yes | No |
| Stenotrophomonas | Yes | No |
| Treponema_2 | Yes | No |
| UBA1819 | Yes | No |
| Varibaculum | Yes | No |
| Victivallis | Yes | No |
| Epulopiscium | No | Yes |
| Neisseria | No | Yes |

[a]ANCOM-BC stands for Analysis of Compositions of Microbiomes with Bias Correction method, and we used model 1, which adjusted for age, education, and infant BMI z-score at 6 months.
[b]CHEU stands for children who are HIV-exposed, uninfected. 'Yes' for structural zero for CHEU and 'No' in CHUU means the bacteria are differentially absent (i.e., absent in CHEU but present in CHUU).
[c]CHUU stands for children who are HIV-unexposed and uninfected. 'No' for structural zero for CHEU and 'Yes' in CHUU means the bacteria are differentially present (i.e., present in CHEU but absent in CHUU).

dyads[15], and the presence of *Peptostreptococcus* in postpartum WHIV may reflect HIV-associated gut dysbiosis and immune changes during the postpartum period, as hormonal and immunologic shifts are common during this time. *Peptostreptococcus* has been implicated in pro-inflammatory signaling and immune modulation, and its presence among postpartum WHIV may indicate an altered gut microbial environment that favors inflammatory pathways[35]. Prior research has not studied the specific connection between HIV in the postpartum period and *Peptostreptococcus*, further studies are needed to confirm the potential relationship.

Finally, our comparison of CHEU and CHUU revealed no differences in α− and β−diversity, similar to previous studies[36]; however, there were differences in the microbiota profile by maternal HIV status based on differential presence/absence of various taxa, consistent with prior findings of dysbiosis[14,37]. In our analysis at 6 months of age, CHEUs had differential presence of *Neisseria* and *Epulopiscium*, a member of the *Lachnospiraceae* family. *Lachnospiraceae* have previously been reported as one of the most abundant bacterial families in CHEU[36]. *Lachnospiraceae* can promote regulatory-T-cell (Treg) differentiation, potentially limiting CHEU's ability to respond to infections[15,38]. There is limited research on *Neisseria* in the gut; however, certain *Neisseria* species have been found to be elevated in inflammatory states and have been linked to gut dysbiosis and immune activation[39]. These findings warrant further study to test whether interventions to modify levels of these bacteria can improve CHEUs' outcomes. For example, potential interventions, such as probiotics or prebiotics, or responsible antibiotic use[40], could be tested on whether they can support growth of beneficial bacteria to improve long-term health outcomes in this population[14].

Our study, with its longitudinal design, intensive and frequent follow-up for microbiota, and paired metabolomic data, offers key insights into how HIV impacts the microbiota and metabolome during pregnancy and postpartum. We also examined how infant microbiota is impacted by maternal HIV exposure status. However, limitations include lack of ART regimen variability, which should be explored in future studies. Further, all the WHIV were already on ART, and our results should be interpreted as differences between WHIV on ART and SN. Additionally, our cross-sectional analyses for some of the analyses further limit causal interpretation. Further, intensive sampling in the post-partum period and in infants for microbiota assessment, along with additional metabolomic profiles at other time points, would also be helpful to better understand the temporal dynamics by HIV status in these populations.

In conclusion, WHIVs have lower α-diversity during pregnancy, with the temporal dynamics consistent over the course of gestation. WHIV also has differences in overall microbiota profile, specific taxa, and metabolome by HIV status during pregnancy. Some of these microbes and metabolites are linked to adverse maternal-infant health outcomes, including those with higher rates observed in WHIV. The microbiota dysbiosis in WHIV persists through the postpartum period, and maternal HIV status also impacts the CHEU microbiota. Future studies should assess whether modulating microbes or metabolites during pregnancy could improve maternal-infant health outcomes and further understand the health impacts of gut dysbiosis in pregnant and postpartum women, and their CHEU.

## Methods
### Study design and population
From June 27, 2016, to December 9, 2019, we conducted a longitudinal cohort study, the PRACHITi study (Pregnancy Associated Changes in Tuberculosis Immunology) in Pune, India[41]. The study focused on pregnant women and was carried out at Byramjee Jeejeebhoy Government Medical College (BJGMC), a tertiary care hospital serving low-income populations and acting as a referral center for HIV care. We enrolled adult pregnant women between the ages of 18 and 40, who

were between 13 and 34 weeks of gestation, as confirmed by early pregnancy ultrasonography. Eligibility criteria for exclusion of women from the study included active tuberculosis, severe anemia, ongoing use of antibiotics for more than 14 days, autoimmune or immunosuppressive diseases, or current use of immunosuppressive medication.

The primary objective of the PRACHITi study was to compare immune responses during pregnancy and postpartum based on HIV status. To address these objectives, pregnant women were enrolled in the study, stratified by their HIV status, based on convenience sampling of participants who met the eligibility criteria within each stratum. The mothers and their infants were followed through 1 year postpartum. For comparison purposes, non-pregnant women were also enrolled for one cross-sectional study visit. For the analysis of pregnancy gut microbiota by HIV status, we included all participating pregnant women (90 WHIV and 154 SN for HIV, during the second and third trimester, and then followed postpartum), and their infants (63 CHEU and 116 CHUU). For the analysis of gut microbiota of non-pregnant women, we have 88 WHIV and 85 SN for HIV from PRACHITi with available stool samples.

To better understand the temporal dynamics of the gut microbiota during pregnancy and how HIV status affects that, as a sub-study of PRACHITi, we conducted frequent stool sampling (every two weeks until 37 weeks of gestational age) in a subset of 71 participants (35 WHIV and 36 SN for HIV) that were enrolled in PRACHITi from 14 to 26 weeks of gestational age (second trimester).

### Ethics

Approval for the study was obtained from the institutional ethics review boards of BJGMC, the Johns Hopkins University, Columbia University, and Weill Cornell Medicine, as well as the Health Ministry Screening Committee of the Indian Council of Medical Research, India. All guidelines required by the US Department of Health and Human Services for Human experimentation were followed. We received written informed consent from all participants. The reporting guideline for observational studies: Strengthening the Reporting of Observational Studies in Epidemiology (STROBE) was followed[42].

### Data collection and procedures

Sociodemographic information (age, education, smoking history) was obtained from all pregnant women at enrollment, either during the second (13–27 weeks of gestation) or third (week 28 onwards) trimester. Clinical and other relevant data were also collected at enrollment (either at the second or third trimester), with additional data collected at the third trimester visit, delivery, and at 6 months postpartum. Newborns were followed from birth through 1 year of age. At these visits, we also measured the maternal mid-upper arm circumference (MUAC) to determine undernutrition (MUAC < 23 cm) during pregnancy, and in select visits assessed gestational diabetes and anemia[43,44]. Blood was collected in the third trimester from all participants, heparin plasma was isolated and extracted, and stored until further assessment. Non-pregnant women were enrolled for one cross-sectional visit, and similar clinical and sociodemographic information was collected. Stool samples were collected at the second trimester, third trimester, and 6 months postpartum visits for the mothers, 6 months of age for the infants, and one time cross-sectionally for the non-pregnant women. For the sub-study, we collected stool samples every two weeks until 37 weeks of gestational age and collected related clinical data at each visit.

### HIV status and sociodemographic/clinical characteristics

The exposure variable in our study was maternal HIV status, categorized as WHIV and SN. Correspondingly, infant HIV status was classified as CHEU for infants born to WHIV mothers and CHUU for those born to SN mothers. For pregnant women enrolled from the antenatal clinic at

Sassoon Hospital in Pune, India, HIV testing is a part of routine care, and information regarding these assays was abstracted from the medical chart. An HIV report within the last 30 days of entry was required for HIV-negative women. For non-pregnant women recruited from the antiretroviral clinic, general medicine, or Obstetrics and Gynecology clinics at Sassoon Hospital, documentation of HIV status was provided through medical charts. If HIV status was unknown, HIV testing and counseling were offered prior to enrollment in the study. We collected the final HIV status of participants at the end of the study, and the HIV status of all participants did not change from 2nd trimester enrollment to 12 months postpartum.

### Gut microbiota

The primary outcome variable for this study was the gut microbiota profile. Stool samples, collected by participants using OMNIgene.GUT kits (DNA Genotek) were utilized to assess gut microbiota composition for pregnant women, non-pregnant women, and infants. Participants collected the stool samples at second trimester (only if enrolled at this timepoint), third trimester, and 6 months postpartum for mothers, and at 6 months of age for infants. For participants enrolled in the sub-study, intensive bi-weekly (i.e., every 2 weeks) stool samples were collected during the second and third trimester. Stool samples were collected by participants at home prior to their next study visit, stored at room temperature (RT) and brought to the clinic at their next visit in RT. These samples were aliquoted and stored at −80 °C prior to shipment on dry ice to the University of Maryland Institute for Genome Sciences (IGS) Microbiome Service Laboratory (https://msl.igs.umaryland.edu/) for sequencing and bioinformatics analysis. Further details are in Supplementary Methods, but briefly, total DNA was extracted from the stool samples using the QIAamp DNA stool extraction kit (QIAGEN, Valencia, CA, USA). Amplicon sequencing of the V3-V4 regions of the 16S rRNA gene was conducted through a two-step PCR process[45]. Sequences were de-multiplexed, primer sequences removed, and DADA2 workflow was used for data processing. Taxonomic assignments were made using the RDP Classifier[46] and the SILVA v138 database[47].

### Metabolomics

From a subset of 100 participants enrolled in PRACHITi (48 WHIV and 52 SN), heparin plasma samples were shipped to Metabolon to conduct unbiased metabolomics. Further details can be found in the Supplementary Methods, but in brief, sample preparation was performed using the automated MicroLab STAR® system (Hamilton Company), with methanol-induced protein precipitation followed by centrifugation. Extracts were aliquoted into four fractions for analysis via ultrahigh-performance liquid chromatography–tandem mass spectrometry (UPLC-MS/MS) under positive and negative ion modes, optimized for hydrophilic, hydrophobic, and ionic compounds using both reverse-phase and HILIC columns. Chromatography was performed on a Waters ACQUITY UPLC system, and mass detection was conducted using a Thermo Scientific Q-Exactive mass spectrometer with a HESI-II source and Orbitrap analyzer at 35,000 resolution. MS analysis alternated between full scans and data-dependent MSn across a 70–1000 $m/z$ range. Raw data was processed for peak detection, compound identification, and quality control.

### Statistical analysis

Descriptive statistics of population characteristics by HIV status were calculated using third trimester samples, which is the timepoint with the most participants. To compare these population characteristics by HIV status, we used Fisher's Exact test for categorical variables and Wilcoxon rank-sum test for continuous variables. $P$-values are from two-sided tests with < 0.05 being considered statistically significant.

For analysis of the sub-study with frequent stool sampling, we first tested whether the microbiota α-diversity was similar (or changing)

within the same pregnant woman from the second to the third trimester by using a paired *t*-test within WHIV, SN, and combined, respectively, comparing second (i.e., mean of all longitudinal second trimester samples) vs. third trimester (i.e., mean of all longitudinal third trimester samples) Shannon α-diversity index. We also conducted linear mixed-effect models (LMM) on longitudinal α-diversities to examine if α-diversity changes across frequent samples from second and third trimesters where we set subjects as random intercepts. This is to understand the temporal dynamics of the α-diversity within WHIV and SN.

We then used combined frequent samples in the sub-study to understand the impact of HIV status on overall microbiota over the course of gestation, also using LMM, with HIV as the exposure variable, adjusting for gestational age with subject as a random effect, on α-diversity outcomes.

Next, we assessed data from the full PRACHITi cohort to assess the associations between maternal HIV status and (1) maternal microbiota during pregnancy (repeated sample analysis combining all second and third trimester samples – analysis 1), (2) maternal microbiota at 6 months postpartum (analysis 2), (3) infant microbiota (i.e. CHEU vs. CHUU) at 6 months of age (analysis 3). In exploratory analyses, we also assessed associations between (4) maternal HIV status and microbiota in non-pregnant women (analysis 4), and (5) pregnancy status (third trimester vs. non-pregnant women) and microbiota in WHIV (analysis 5).

For these various analyses in the full cohort, we analyzed microbiota data as α-diversity, β-diversity, and taxa abundance. First, we assessed microbiota α-diversity by comparison groups. For the first comparison of maternal HIV status and maternal microbiota during pregnancy (analysis 1), we conducted LMMs with subject as a random effect, α-diversity as outcome, and HIV status as exposure variable, and adjusting for maternal age, education, and MUAC, for model 1. To determine the association between the exposure variable (i.e., HIV status (analysis 2–4) or pregnancy status (analysis 5)) and α-diversity for the above comparisons, we conducted *t*-tests for model 0 and multivariable linear regression for model 1. In these multivariable analyses, we adjusted for age (years), education (none to high school or post high school to postgraduate), and anthropometrics (i.e., mid-upper arm circumference (MUAC) in cm for mothers and body mass index Z-scores for infants) based on a directed acyclic graph (DAG).

Next, we compared overall microbiota profile by HIV status in the full cohort using PERMANOVA on β-diversity (based on Bray-Curtis distance) for the above comparisons[48]. For the pregnancy PERMANOVA analysis (analysis 1), we conducted separate analysis for each trimester and only used one sample (based on protocol-assigned second or third trimester sample) per trimester.

Finally, we conducted analysis comparing individual taxa at the genus level with above-defined comparison groups in the full cohort using the ANCOM-BC2 method, version 2.10.1, with pseudocount of 0.1 and structural zero as true[49]. For analysis 1, the ANCOM-BC2 model was fitted to compare the bias-corrected log absolute abundance of each taxon between WHIV and SN groups over longitudinal samples from the second and third trimesters, where the bias correction is due to sampling fraction, the group status is an explanatory variable, and the subject is treated as a random effect, along with additional covariates included in the model. The log fold change of the bias-corrected log absolute abundance by the group status was calculated for each taxon. The p-values for taxa were adjusted for multiple comparisons using either the BH method to calculate adjusted p-values (which is the focus of our interpretations) or the Bonferroni method to calculate 95% confidence intervals (which is being used primarily for visualization purposes). In addition to differentially abundant taxa, we also report structural zero taxa as differentially present/absent bacteria (i.e., differentially present in one group but not another). Of note,

differentially abundant taxa and differentially present/absent taxa refer to distinct analytical approaches used in ANCOM-BC to deal with each type of taxa. However, both are considered together for interpretation of differences between study groups. We analyzed two different ANCOM-BC models: crude (model 0 with no covariates) and adjusted model 1 with the same covariates, maternal age, education, and MUAC, as in analysis with α-diversity. Microbiota samples with less than 1000 total reads were removed from all microbiota analyses. Analogously, for analyses 2–5, we fit ANCOM-BC2 using cross-sectional data and reported similar quantities of log fold changes and *p*-values for taxa to compare the two groups. Further, using a linear mixed model (LMM), we assessed the log ratio of *Bacteroides* to *Prevotella* during pregnancy as the outcome variable, HIV status as the exposure variables and subject as random effect and generated an ANCOM-BC genera table with pseudocount of 0.1.

For our analysis of metabolomics data, we normalized each metabolite by dividing the metabolite values with its median. We then imputed metabolite below the detection limit by the minimum divided by two for each metabolite. The values were then natural log transformed. We conducted logistic regression with HIV status as the outcome, and metabolite measures as the predictor, adjusting for maternal age, education, and MUAC, the same covariates in the microbiota analysis. Multiple comparisons were corrected using FDR with BH adjustment. In the sub-sample with both microbiota and metabolome data, we assessed correlations between microbes (with raw p-value ≤0.1 in ANCOM) and metabolites (BH-adjusted *p*-value < 0.05 using a *t*-test). All statistical analyses were two-sided and performed using R (version 4.5.1).

### Reporting summary

Further information on research design is available in the Nature Portfolio Reporting Summary linked to this article.

## Data availability

The raw sequencing microbiome data and associated metadata that support the findings of this study have been deposited to the sequence read archive repository under the BioProject Number PRJNA1224750 (http://www.ncbi.nlm.nih.gov/bioproject/1224750). Mass spectrometry metabolomics data is available in the Metabolomics Workbench repository[50] under Study ID: ST004581 and through https://doi.org/10.21228/M8MP24.

## Code availability

Analysis codes and data files used to generate the results are publicly available on GitHub (https://github.com/tianwang-wow/Pregnancy_microbiome_in_HIV) with the following DOI in Zenodo[51]: https://doi.org/10.5281/zenodo.18369924.

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

## Acknowledgements

The authors thank the study participants for their time and contributions, as well as the study staff who meticulously collected detailed data. Research reported in this publication was supported by the Eunice Kennedy Shriver National Institute of Child Health and Human Development of the National Institutes of Health under award number R00HD089753 to RS, R01HD081929 to AG, as well as NIAID (K23AI129854 to JSM). Additional support for this work was the NIH-funded Johns Hopkins Baltimore-Washington-India Clinical Trials Unit for NIAID Networks (UM1AI069465 to AG). J.R., M.S.H., and J.B.H. were supported by the National Institute of Nursing Research, of the National Institute of Health under award number R01NR015495. The content is solely the responsibility of the authors and does not necessarily represent the official views of NIH. The authors also acknowledge in-kind support from Persistent Systems.

## Author contributions

J.M. conducted the data analysis and wrote the primary version of the manuscript. T.W. conducted the primary data analysis and contributed to the writing and interpretation of findings. J.S.M., S.N., R.B., A.K., and A.G. led the parent study and contributed to the design, implementation, and interpretation of this study. M.S., B.M., J.B.H., K.G.G., S.W., and J.R. contributed to the design, implementation, analysis, or interpretation of this study. M.A. contributed to the study design and interpretation and led the data collection. V.K., P.D., and M.S.H. contributed to laboratory data collection and review of this manuscript. R.S. led the conceptual design and contributed to data analysis, manuscript writing, and review. All authors contributed to manuscript writing, have approved the final manuscript, and agreed to publication.

## Competing interests

J.R. is co-founder of LUCA Biologics, a biotechnology company focusing on translating microbiome research into live biotherapeutic drugs for women's health. For the remaining authors, none were declared.
