## [Transparent Peer Review file · Nature Communications]

Impact of maternal HIV infection on gut microbiota and metabolome of maternal-infant populations

Corresponding Author: Dr Rupak Shivakoti

Version 0:

Reviewer comments:

Reviewer #1

(Remarks to the Author)
Please see the attached report

(Remarks on code availability)

Reviewer #2

(Remarks to the Author)
Mandell et al examined how maternal HIV infection impacts gut microbiota composition across second and third trimesters in pregnancy as well as the systemic metabolome. They also assessed how HIV exposure impacts gut microbiota in infants. Overall, the manuscript is well written, but there are areas that can be improved. It would be helpful to the reader if the analyses can be simplified and time points clearly stated to allow the reader to easily follow the manuscript. Authors should also integrate the plasma metabolome with the microbiota to strengthen insights into functional pathways impacted by HIV.

Major comments:

Metabolomics analyses was only done in the third trimester while microbiota analyses were done in the second and third trimesters. Can authors also look at the metabolome in the second trimester if samples are available?
As presented, the work is very descriptive. It would help to link the changes in microbiota to the metabolites. For example, do we know which of the metabolites measured in plasma are microbe versus host derived? These analyses would strengthen the connection between changes observed in gut composition and the systemic metabolome.

Minor comments:

How often was sampling done from the full cohort?

Provide refence for the STROBE guidelines

Is there any reason why you have two multivariable models (1 and 2 which differ in covariates) for analyses? This just complicates the analyses making it difficult for the reader. Authors should show the univariable (unadjusted) and the multivariable (adjusted for all covariates)

For taxa filtering, what does an average of 0.0005% across samples mean? Is this mean abundance? median abundance?

What % of samples did the taxa have to be present on to be kept?

Please include equations for the various linear mixed models.

Since authors included women with detectable HIV viral load, is there a difference in microbiome between those with and without detectable VL?

Discuss other papers that have done similar work ie PMID 24034618; 38230936; 39707483; 30657007

Generally, in the figures where you are doing differential abundance testing, please show directionality on the figure itself so it is easy to visualize which bacteria are abundant in what group. Unless the plots are only showing what is up or down within one group?

What method was used to analyze the differentially abundant metabolites across groups?

What exactly is Figure 3 showing? The legend says metabolites but the results section say pathways. Also, the legend says bacteria yet this is supposed to be metabolites?

Sphingomyelins are metabolites, if pathway do you mean sphingomyelin metabolism?

(Remarks on code availability)

Link has accession numbers of deposited 16S raw fastq sequences from the study.
There is no analysis code provided.

Version 1:

Reviewer comments:

Reviewer #1

(Remarks to the Author)

This is a very well done manuscript. The revision addresses all my concerns and comments. I have no further comments.

(Remarks on code availability)

Reviewer #2

(Remarks to the Author)

Authors have addressed all my concerns.
Here are some minor additional comments:

Is the full cohort made of 245 or 242? The number in the abstract is different from the one in text.

The manuscript has several sections saying "data/results not shown". Could this maybe be shown as supplementary or removed if not important?

Is there a reason why in the LMM models, you use different methods for p value adjustment? In others you use Bonferroni and BH in others.

You say in line 314 that WHIV had a lower alpha diversity compared to controls, but the p value is 0.12. You mention that Chao1 and Fisher was significant, but that data is not shown.

Is the heatmap in Supp Fig 2 complete? You mention 17 taxa but only 3 are shown on rows on the heatmap. The metabolites on columns are also absent and the grouping information on the heatmap is missing.

Line 625–628, the discussion of the two taxa here that were differentially abundant in the second and third trimester appear misplaced. This section is for differentially abundant taxa in the postpartum period.

Include references in the discussion, line 723

Since you mention the differentially present/absent taxa in the results (and these are different from differentially abundant taxa), could you please discuss in a bit more detail what the implications of such taxa might be for WHIV or their CHEU?

(Remarks on code availability)

December 24, 2025

We thank the reviewers for the constructive and helpful comments on our manuscript titled “Impact of maternal HIV infection on gut microbiota and metabolome of maternal-infant populations.” We have revised our manuscript addressing each comment from the reviewers as shown in detail below with a point-by-point response. Please note that the number lines refer to the highlighted version of the manuscript.

Reviewer Comments:

Reviewer #1:

1. **Models:** The authors interpreted their data using three different statistical models. Model 0 is the crude model unadjusted for any covariates, Model 1 adjusted for age, education, and anthropometrics, and Model 2 further adjusted Model 1 for smoking status, gestational diabetes (GDM) and anemia. While, I can understand Model 1, but to me Model 2 does not seem appropriate. Please note that HIV condition can disrupt the gut microbiome which in turn can lead to anemia or HIV may have a direct effect leading to anemia. Thus, anemia is potentially a collider and not a confounder. Anemia does not affect HIV status and not sure if it causes changes in the microbial ecology. It will be helpful, if the authors draw a DAG to explain why they should adjust for anemia. For the same reasons, I am not sure about GDM either. Furthermore, the two groups don't seem to be significantly different (Table 1). Since smoking is not very common in this population (Table 1) and the two groups don't appear to be significantly different, I think it is better not to adjust for it in the model. It is sufficient to present results from Model 1 only.

Response: Thank you for pointing this out. Based on a similar suggestion from Reviewer #2, we have removed model 2 from all tables, figures and the manuscript. We now only focus on model 1 as we agree that anemia and GDM do not have a direct effect on HIV and could be colliders rather than confounders.

2. **Sexual activity and substance use.** Major contributors to HIV infection are sexual activity and substance use. It would be helpful if the authors provided information on number of sexual partners the women had and whether any of the partners are known to be people with HIV. If such data are available, then summarize these data in Table 1. Also, cross-tabulate the HIV status with number of sexual partners. This can be relegated to supplementary text. Similarly, it would be useful to provide information on substance use and needle sharing.

Response: Thank you for this suggestion. We have data on substance use (drugs and alcohol), and in a subset data on number of sexual partners and HIV partner exposure. This has been added to **Table 1**, and a supplementary table on HIV status and the number of sexual partners has also been added. **Supplementary Table 3** has the cross-tabulation of maternal HIV status at third trimester by number of sexual partners.

Lines 264-268: “*The median number of sexual partners in a participant’s lifetime was 1 for WHIV and SN. WHIV had no partners newly diagnosed with HIV (Table 1), and a cross-tabulation of maternal HIV status at third trimester by number of sexual partners in lifetime is shown in Supplementary Table 3. Of those with data, there were no participants who used drugs or had alcoholic drinks in the past 2 years (Table 1).*”

3. **Spaghetti plots:** While the spaghetti plots tracking the alpha diversity over the gestational age between second and third trimesters is interesting to visualize, I find them to be difficult to interpret. Secondly, in a normal pregnancy (without HIV infection or any other health condition), the alpha diversity as well as the microbial compositions are expected to change during the course of pregnancy (Kore et al., Cell, 2012). The alpha diversity is expected to decrease with gestational age in a normal pregnancy (Mesa et al., Nutrients 2020). In view of this, it would be interesting to test if this hypothesis is true by HIV status. The authors could use the software package CLME (Constrained Linear Mixed Effects) to test whether the mean alpha diversity is decreasing within each HIV group. The CLME software is designed for testing such hypotheses.

Response: Thank you for the suggestion to test whether the mean alpha diversity is decreasing over time, and whether there are differences by HIV status. We now present data in Supplementary Table 4 that addresses this point. Paired t-test of the mean alpha diversity of second trimester samples and mean of third trimester samples did not show a decreasing trend of alpha diversity with gestational age in analyses of the combined population (WHIV+SN) as well as analyses within each group (WHIV and SN). We also present data from a LMM to show the coefficient of gestational age in WHIV and SN. This suggests stable alpha diversity during this period of second and third trimester in both groups. We have added the following text in the method section lines 715-727 to detail the methods:

Lines 715-727: “*For analysis of the sub-study with frequent stool sampling, we first tested whether the microbiota α -diversity was similar (or changing) within the same pregnant woman from the second to the third trimester by using a paired t-test within WHIV, SN, and combined, respectively, comparing second (i.e. mean of all longitudinal second trimester samples) vs. third trimester (i.e. mean of all longitudinal third trimester samples) Shannon α -diversity index. We also conducted linear mixed-effect models (LMM) on longitudinal α -diversities to examine if α -diversity changes across frequent samples from second and third trimesters where we set subjects as random intercepts. This is to understand the temporal dynamics of the α -diversity within WHIV and SN.*

We then used combined frequent samples in the sub-study to understand the impact of HIV status on overall microbiota over the course of gestation, also using LMM, with HIV as the exposure variable, adjusting for gestational age with subject as a random effect, on α -diversity outcomes.”

We also added the following lines in the results and results section lines 278-299 and also improved our explanation of the spaghetti plot in the legends for the reader to better understand these figures:

Lines 278-299: “*We assessed whether microbiota α -diversity, Shannon index specifically, was similar between the second and third trimester using the 71 pregnant women with frequent sampling. Using a paired t-test comparing mean α -diversity from all the second trimester*

samples to that of the mean of all third trimester samples (Supplementary Table 4), we observed no significant differences across all 71 pregnant women or by HIV status.

Using the same frequent samples, we also used linear mixed models (LMM) with longitudinal α -diversity measures through second and third trimesters as the outcome and gestational age (in weeks) as the explanatory variable, and treated subject as a random effect to examine whether α -diversity changes over time in pregnancy within SN and WHIV, separately. The coefficient of gestational age within SN is 0.005 ($p=0.30$) and within WHIV is <0.001 ($p=0.93$). This suggests that α -diversity does not change over time during the second and third trimester within either group. We observed similar results using other α -diversity indices (e.g. Chao1 and Fisher (data not shown)).

With the conclusion that microbiota does not differ between second and third trimesters, we then tested whether α -diversity differs by HIV status using all longitudinal samples of the same pregnant woman ($n=71$) from the second and third trimesters with LMMs, where we have repeated α -diversity measures as outcomes and HIV status as the predictor. We observed that WHIV had significantly lower α -diversity ($p=0.015$) than SN women between the second and third trimester (Figure 1). We observed similar results with other α -diversity indices (e.g. Chao1 and Fisher (data not shown)). In summary, we observe that WHIV had lower α -diversity than SN pregnant women, with a consistent relationship through the second and third trimester of gestation.”

Of note, the Koren paper that showed changes was comparing the first trimester to the third trimester, while our data is focused on the second and third trimester with consistent results from another paper studying this time period (PMID: 26283357). We have added the following text in lines to clarify this difference:

Lines 497-502: *“The temporal dynamics of gut microbiota α -diversity, however, was consistent through gestation and did not differ by HIV status. Similar results were observed in populations without HIV, where the gut microbiota α -diversity did not change over gestation¹⁶. Although some studies have noted decreasing α -diversity as pregnancy progresses, there are important differences in the time-point of comparison data (i.e. notable differences between the first and third trimester while ours compared second and third trimester)^{33, 34}.”*

4. **Figure 2 (and Supplementary tables 2 and 3):** As noted above in my point #2, the gut microbiota are continuously evolving during the course of pregnancy. In view of that, it does not seem appropriate to combine data from trimesters 2 and 3 and compare the HIV groups as the primary analysis (Figure 2) and treat differential abundance analysis of individual trimesters as secondary analysis (Supplementary tables 2 and 3). Also, it appears some taxa are reported at the family level and others at genus level. Is there a reason to do so? Also, I was surprised to see very few taxa to be differentially abundant, whereas a large number of taxa are differentially abundant at postpartum.

Response: Thank you for the feedback. We have addressed this in the response above by showing that the gut microbiota is stable during the second and third trimester. This motivates us to keep the second and third trimester samples together in our analysis, as it is justified by these new analyses, consistent with our original approach, and helps with the power of the analyses.

Related to taxa reporting at the family level and genus level, we want to note that all of the taxa analyzed and reported for ANCOM-BC are classified at the genus level according to SILVA phylogeny. We used the SILVA v132 dataset to assign taxonomy to ASVs. In the results section, we refer to certain families only to group multiple significant genera that belong to the same family, but the analysis itself was performed at the genus level.

Based on reviewer comment on the differences between pregnancy (few taxa significant) and postpartum (larger number of taxa significant), we have reanalyzed our data using the latest version of ANCOM-BC. This version is more conservative, especially in how it analyzes pseudocounts, resulting in fewer taxa that are significant. As a result, while the pregnancy results remain consistent, there are changes to both the postpartum and infant results and we have updated these sections accordingly in the results and discussion section.

5. **Bacteroides and Prevotella:** Many *Bacteroides* species are commensal gut bacteria and some of them play an important role in the production of short chain fatty acids. On the contrary *Prevotella* species are known to be proinflammatory and reported in the literature to have increased abundance in people with HIV. Hence, I was surprised that these taxa were not differentially abundant in this pregnancy cohort. It will be interesting to see the log ratio of *Bacteroides* to *Prevotella* is significantly different by HIV status in this population.

Response: Thank you for this comment. As suggested, we further looked at the mean abundance of each taxa and the log ratio of *Bacteroides* to *Prevotella* by HIV status. As noted by the reviewer, *Prevotella* had a greater mean log abundance in WHIV compared to SN (0.0002 vs. 0.0001) but this association was not statistically significant in FDR-adjusted analyses. However, *Bacteroides* mean log abundance was also higher, but not statistically significant, in WHIV compared to SN (0.08 vs. 0.05).

Related to the log ratio, the mean abundance of *Bacteroides* was higher than the mean abundance of *Prevotella* in both WHIV and SN, with a higher log ratio of *Bacteroides* to *Prevotella* in WHIV (7.39) compared to SN women (6.59). Further, we fit a linear mixed effect model: $\log(\textit{Bacteroides}) - \log(\textit{Prevotella}) \sim \text{HIV status} + \text{subject (random effect)}$. When we used the ANCOM-BC bias corrected genera table, we observed higher log ratio of *Bacteroides* to *Prevotella* in WHIV compared to SN ($p=0.08$). Thus, this is in the reverse direction as hypothesized by the reviewer.

We added a line about this analysis in the methods:

Lines 768-771: “Further, using a linear mixed model (LMM), we assessed the log ratio of *Bacteroides* to *Prevotella* during pregnancy as the outcome variable, HIV status as the exposure variables and subject as random effect and generated an ANCOM-BC genera table with pseudocount of 0.1.”

We also noted these findings in the results section lines 360-370:

“Prior studies have noted increased levels of pro-inflammatory *Prevotella* in HIV²⁷, while *Bacteroides*, which includes many commensal gut bacteria important in SCFA production²⁸, may

be lower. We specifically assessed these two bacteria to further understand their levels in WHIV during pregnancy and to test our hypothesis on whether the ratio of *Bacteroides* to *Prevotella* would be lower in WHIV. However, mean abundance of *Prevotella* (0.0002 vs. 0.0001 in WHIV and SN) and *Bacteroides* (0.08 vs. 0.05 in WHIV and SN) were both higher in WHIV compared to SN, although this association was not statistically significant in FDR-adjusted ANCOM-BC2 analyses on genus level taxa. We further conducted LMM with log ratio of *Bacteroides* to *Prevotella* in longitudinal samples as the outcome, HIV status as the exposure and subject as the random intercept and observed higher log ratio of *Bacteroides* to *Prevotella* in WHIV compared to SN ($p=0.08$), which is not consistent with our hypothesis on *Bacteroides* to *Prevotella* ratio.”

6. **Figure 4:** These are interesting postpartum results. I was surprised to see *Akkermansia* to be the most differentially abundant taxa with higher abundance in women with HIV. It is well-known that *Akkermansia muciniphila* plays an important role in the gut health by contributing to the production of short chain fatty acids. I would have expected it to have higher abundance in women without HIV. It will be useful to perform CLME analysis, by HIV status, to see if the alpha diversity decreased from 2nd trimester to third trimester and then increased 6 months postpartum. Thus, a “U” shaped response in mean alpha diversity.

Response: Thank you for this suggestion. We created side-by-side boxplots by HIV status for alpha diversity for the second trimester, third trimester and 6 months postpartum (**Supplementary Figure 2**). Our results did not show a “U” shaped response in mean alpha diversity for the Shannon index. For both WHIV and SN samples, 2nd and 3rd trimester mean alpha diversity remained constant and decreased slightly at postpartum. The WHIV mean alpha diversity remained lower than the SN mean alpha diversity for all 3 timepoints. We have added the following text in lines 427-429 of the results section to detail this comparison:

*“Across the three time periods by HIV status, α -diversity remained unchanged from second to third trimester but decreased slightly at postpartum (**Supplementary Figure 1**).”*

Related to *Akkermansia* results, after using the more updated conservative ANCOM-BC version mentioned above, the abundance of *Akkermansia* was not significantly different by HIV status.

7. **Metabolomics:** Although it appears the authors obtained data from both positive and negative ion channels, it is not clear if they analyzed them separately (which is the preferred approach) or did they combine them into one? Also, in the methods section, they indicate that they adjusted for the “same covariates as in the microbiota analysis”. However, it is not clear to me if the results presented in Supplementary Table 4 are based on Model 1 or Model 2? I suggest presenting results only from Model 1.

Response: Thank you for pointing this out. For our metabolomics analysis, we followed the standard procedures used by Metabolon (the lab which conducted the metabolomics) and in their various collaborations using this platform. Metabolon uses multiple arms (both negative positive ion channels) (LC-MS/MS method) within a platform to try to maximize biochemical coverage. Data from all arms are then combined into a single dataset and analyzed together. Although certain metabolites can be detected across multiple arms of the platform, data from only one arm is reported for each metabolite and is selected by them based on sensitivity and reproducibility parameters, which is consistent across all samples. Each metabolite is identified by matching the

ion chromatographic retention index, accurate mass, and mass spectral fragmentation patterns with a reference library created from standard metabolites analyzed with identical instrumental procedures. PMID: 32445384 explains these methods in detail as well. Of relevance to our data, the global discovery platform that we used is an untargeted method that uses relative quantification. Thus, statistical comparisons are only made between study groups (i.e. WHIV and SN), and not between individual metabolites.

Regarding the models, we have now made it clearer in both the manuscript and **Supplementary Table 5-8, and 10** that we are only presenting results from model 1.

Lines 763-765: *“We analyzed two different ANCOM-BC models: crude (model 0 with no covariates) and adjusted model 1 with the same covariates, maternal age, education and MUAC, as in analysis with α -diversity.”*

8. **Table 1:** shows the distributions at 3rd trimester. Did the HIV status of any participant change from 2nd to 3rd trimester of pregnancy. This might be important to consider because the authors combined 2nd and 3rd trimesters together. Is there a similar table available for 2nd trimester? What about infants?

Response: Thank you for this suggestion. The HIV status did not change for any participant from 2nd trimester to 3rd to postpartum and mentioned this in the results at lines 671-673:

“We collected the final HIV status of participants at the end of the study, and all participants HIV status did not change from 2nd trimester enrollment to 12 months postpartum.”

Also, we have now added a table 1 for 2nd trimester and 3rd trimester into the supplementary materials. They are **Supplementary Table 1 and 2** and referred to in the text lines 268-269:

*“Study population characteristics for mothers in the second trimester and their infants can be found in **Supplementary Table 1 and 2.**”*

9. **Breastfeeding:** Given that this is a low-income population, due to affordability, I expected almost all babies to breastfed, but it appears the opposite is the case (exclusively breastfed = 60 and not exclusively breastfed = 116).

Response: Thank you for pointing this out. In our manuscript, we referred to exclusive breastfeeding. However, when it comes to breastfeeding in general, in fact, 86% of infants are breastfeeding at 6 months postpartum. Given their age of 6 months, it is common for complementary feeds to be introduced in this setting, despite recommendation to exclusively breastfeed through 6 months, which is why we decided to use the exclusive breastfeeding variable instead. We have now added the following text to specify the breastfeeding rates in addition to the exclusive breastfeeding % as follows in lines 466-468:

“In our cohort, 86% of infants were being breastfed, and 34% exclusively breastfed at 6 months of age, with a slightly higher proportion of CHEUs (36%) exclusively breastfed at 6 months than CHUUs (33%).”

10. **Measures of alpha diversity:** While it is impressive to consider so many different measures of alpha diversity, I think it is sufficient to consider just one measure of their choice and mention that other measures also considered with similar results (data not shown).

Response: Thank you for your suggestion. We agree with your suggestion and have decided to only show and discuss the Shannon diversity index in our manuscript and figures, while mentioning that similar results were observed for the other indices.

11. **Non-pregnant women and third trimester:** I am not sure why the authors compared the non-pregnant women with third trimester women. If anything, I would have like to see the results by comparing post-partum data with non-pregnant women data by HIV status. This would tell us whether the postpartum gut ecology is closer to that of non-pregnant women, in the respective HIV groups.

Response: Thank you for your suggestion. We conducted the analysis between non-pregnant WHIV women and third trimester WHIV women because we wanted to understand the impact of pregnancy among those with HIV on the microbiota. A line has been added in the manuscript to clarify this (lines 366-369). We also conducted the suggested analysis between non-pregnant WHIV women and postpartum WHIV women and found no significant differences in alpha diversity, beta diversity and ANCOM-BC (bacteria abundance) analyses. This would suggest that the postpartum gut ecology is closer to that of non-pregnant women's gut microbiome. A line describing this has also been added to the manuscript (lines 435-438).

Lines 378-381: *“Further, in additional exploratory analyses, we compared the microbiota profile by HIV status in non-pregnant women and conducted another analysis comparing the microbiota between pregnant WHIV and non-pregnant WHIV to understand the impact of pregnancy among those with HIV (discussed further in **Supplementary Results**).”*

Lines 446-449: *“In an additional analysis (results not shown), we also observed that the gut microbiota profile of postpartum WHIV more closely resembles that of non-pregnant WHIV, as no significant differences between groups were observed.”*

Reviewer #2:

Major comments:

1. Metabolomics analyses was only done in the third trimester while microbiota analyses were done in the second and third trimesters. Can authors also look at the metabolome in the second trimester if samples are available?

Response: We appreciate this comment. However, since our shipment of third trimester plasma samples to the US, the Indian government regulations related to sample shipment out of the country have changed. They only allow limited samples for quality control purposes to be shipped out of country. Doing the metabolomics analyses in India would also provide challenges related to comparisons of the number and types of metabolites identified, as the third trimester samples have been run with the Metabolon platform in the US. Along with

other logistical challenges – e.g. third trimester had the most samples, hence was our focus, while second trimester had fewer samples, some of which have been used for other assays – obtaining second trimester metabolomics data is not feasible at this time. We agree that it would have been helpful to obtain second trimester data, and have specified this as a limitation our study in lines 593-595:

“Further, intensive sampling in the post-partum period and in infants for microbiota assessment, along with additional metabolomic profiles at other time points, would also be helpful to better understand the temporal dynamics by HIV status in these populations.”

2. As presented, the work is very descriptive. It would help to link the changes in microbiota to the metabolites. For example, do we know which of the metabolites measured in plasma are microbe versus host derived? These analyses would strengthen the connection between changes observed in gut composition and the systemic metabolome.

Response: Thank you for this helpful suggestion. We conducted analyses to link the significant genera in our combined second and third trimester microbiome analysis with the significant metabolites in our metabolomics analysis. We have added a figure and supplementary table to show the multi-omics relationship between the microbiota and the metabolome (**Figure 3b, Supplementary Table 9, and Supplementary Figure 2**).

We have also added a few lines in the results and discussion that mention these findings:

Lines 406-420: *“Out of 100 pregnant women with plasma metabolome measures from the third trimester, 88 also have third trimester microbiota data. Using this subset of 88 pregnant women, we repeated the ANCOM-BC analysis and identified 14 differentially abundant genera with unadjusted p-values ≤ 0.1 , and repeated a two-sample t-test and identified 87 significant metabolites. We then assessed correlations between abundance of the 14 microbes and metabolomic measures of the 87 significant metabolites (**Figure 3b and Supplementary Table 9**). We found taxa *Fusobacterium* had the strongest correlations with several metabolites, including positive correlations with 2-hydroxyglutarate and phenol glucuronide, and inverse correlations with hydroxypalmitoyl sphingomyelin (d18:1/16:0(OH)), glycosyl ceramide (d18:1/20:0, d16:1/22:0), palmitoyl sphingomyelin (d18:1/16:0), and sphingomyelin (d18:2/23:1) (**Figure 3b and Supplementary Table 9**). Taxa *Ruminococcaceae_UCG-003* was positively correlated with metabolite 5alpha-pregnan-3beta,20alpha-diol monosulfate and pregnenolone sulfate (**Figure 3b and Supplementary Table 9**). Additionally, the correlation of the 17 differentially present/absent structural zero genera from our ANCOM-BC analysis with the 87 significant metabolites is presented in **Supplementary Figure 2**.”*

Lines 534-548: *“In our multi-omics analysis, there were several notable correlations between specific microbes and metabolites that were significantly different between WHIV and SN. The strongest positive correlation was observed between *Fusobacterium* and 2-hydroxyglutarate, with both having higher levels than SN. HIV infection is known to affect glutamate and glutamine metabolism, which subsequently impacts immunity and neurotoxicity⁴⁰. Increase in *Fusobacterium* and 2-hydroxyglutarate within WHIV might reflect this altered metabolism since *Fusobacterium* is known to ferment glutamate through the hydroxyglutarate*

pathway⁴². Further research is needed to better understand these relationships, and their impact in pregnancy. The strongest negative correlations were observed between *Fusobacterium* and several sphingomyelin and related metabolites. *Fusobacterium* contain sphingolipids⁴³, thus higher levels of *Fusobacterium* could mean increased use of sphingomyelin by the bacteria and thus lower circulating levels of sphingomyelin. Alternatively, these and other relationships (e.g. positive correlation between *Ruminococcaceae_UCG-003* and 5alpha-pregnan-3beta,20alpha-diol monosulfate) could reflect indirect relationships between microbes and metabolites due to a shared pathway (e.g. inflammation^{39, 44, 45}) impacted by HIV.”

Minor comments:

1. How often was sampling done from the full cohort?

Response: Stool sampling was done at the second trimester (for those who were enrolled at this timepoint), third trimester, and 6 months postpartum for mothers, and at 6 months of age for infants. We have now added this specific information in the methods section along with how participants store their samples.

Lines 679-684: “*Participants collected the stool samples at second trimester (only if enrolled at this timepoint), third trimester, and 6 months postpartum for mothers, and at 6 months of age for infants. For participants enrolled in the sub-study, intensive bi-weekly (i.e. every two weeks) stool samples were collected during the second and third trimester. Stool samples were collected by participants at home prior to their next study visit, stored at room temperature (RT) and brought to the clinic at their next visit in RT.*”

2. Provide refence for the STROBE guidelines

Response: We have now added a reference for the STROBE guidelines at line 643.

3. Is there any reason why you have two multivariable models (1 and 2 which differ in covariates) for analyses? This just complicates the analyses making it difficult for the reader. Authors should show the univariable (unadjusted) and the multivariable (adjusted for all covariates)

Response: Thank you for this comment and we agree with it. Based on previous comments from Reviewer #1, we have decided to only use multivariable model 1 in our analyses which adjusts for age, MUAC, and education. Please see response #1 under reviewer #1 comments.

4. For taxa filtering, what does an average of 0.0005% across samples mean? Is this mean abundance? median abundance? What % of samples did the taxa have to be present on to be kept?

Response: Thank you for this comment. We have removed this criteria in the latest ANCOM analyses as the default setting of ANCOM removes taxa that present in less than 10% of the samples.

5. Please include equations for the various linear mixed models.

Response: We have now included the equations for the linear mixed models in the methods section.

Lines 737-740: *“For the first comparison of maternal HIV status and maternal microbiota during pregnancy (analysis 1), we conducted LMMs with subject as a random effect, α -diversity as outcome and HIV status as exposure variable, and adjusting for maternal age, education, and MUAC, for model 1.”*

6. Since authors included women with detectable HIV viral load, is there a difference in microbiome between those with and without detectable VL?

Response: Thank you for this question. Based on your suggestion, we completed an analysis to assess microbiota differences between third trimester WHIV with detectable viral load and WHIV with undetectable viral load. Beta diversity and taxa abundance were not different by group. However, alpha diversity was significant with the Shannon index higher among pregnant WHIV with detectable viral load compared to pregnant women WHIV with undetectable viral load. We added a line with these results.

Lines 373-377: *“In an exploratory analysis, we also tested whether there was a difference in the third trimester microbiota of WHIV with detectable viral load (VL) as compared to WHIV with undetectable VL. We did not observe differences in β -diversity or taxa relative abundance but observed that the α -diversity Shannon index was higher among WHIV with detectable VL than undetectable VL (data not shown).”*

7. Discuss other papers that have done similar work ie PMID 24034618; 38230936; 39707483; 30657007

Response: Thank you for suggesting these papers. We incorporated the papers we felt aligned best with our results in the discussion section (PMID: 39707483 and 30657007).

Lines 564-567: *“Finally, our comparison of CHEU and CHUU revealed no differences in α - and β -diversity, similar to previous studies⁴⁹; however, there were differences in the microbiota profile by maternal HIV status based on differential presence/absence of various taxa, consistent with prior findings of dysbiosis^{14, 50}.”*

Lines 570-571: *“Lachnospiraceae have previously been reported as one of the most abundant bacterial families in CHEU⁴⁹”*

8. Generally, in the figures where you are doing differential abundance testing, please show directionality on the figure itself so it is easy to visualize which bacteria are abundant in what group. Unless the plots are only showing what is up or down within one group?

Response: Thank you for this comment. The figures of differential abundance testing show log fold change dots and confidence interval bars for the WHIV group only. If the bar is on the right

side of zero, there is high abundance of that bacteria in the WHIV group compared to SN group. If the bar is on the left side of zero, there is lower abundance of that bacteria in the WHIV group compared to SN group. We added some clarification to the legends of these figures in both the main and supplementary files for readers to better understand these plots.

9. What method was used to analyze the differentially abundant metabolites across groups?

Response: We used logistic regression by using a binomial generalized linear model (GLM) with a Wald z-test treating HIV status as the outcome, and metabolite measures as the predictor and adjusting for age, education, and MUAC, to analyze the differentially abundant metabolites across groups. We have now added this at lines 774-777 in the manuscript, and we have also added it in the legend of **Supplementary Table 8**.

Lines 774-777: *“We conducted logistic regression with HIV status as the outcome, and metabolite measures as the predictor, adjusting for maternal age, education, and MUAC, the same covariates in the microbiota analysis.”*

10. What exactly is Figure 3 showing? The legend says metabolites but the results section say pathways. Also, the legend says bacteria yet this is supposed to be metabolites?

Response: Thank you for pointing this out. Figure 3 is showing the significant metabolites in the third trimester metabolomics analysis. We have now updated Figure 3 legend to say “metabolites” as that is what our plot is showing as well as the image.

11. Sphingomyelins are metabolites, if pathway do you mean sphingomyelin metabolism?

Response: Thank you for your comment. We have now changed this sub-pathway name in our main manuscript text to be “sphingomyelin metabolism” as well as any other ones that would arise confusion.

Lines 394-402: *“Among the 40 metabolites elevated in WHIV, the significantly enriched metabolites were from the following sub-pathways: fatty acid metabolism, N-acyl amino acids, pyrimidine metabolism, purine metabolism, corticosteroids, and food component/plant (Supplementary Table 8). Among the 47 metabolites lower in WHIV, these were from sub-pathways including sphingomyelin metabolism, progestin steroids, secondary bile acid metabolism, dihydrosphingomyelin metabolism, hexosylceramides, and androgenic steroids (Supplementary Table 8).”*

12. Link has accession numbers of deposited 16S raw fastq sequences from the study. There is no analysis code provided.

Response: We have now included our analysis code for your reference in lines 815-817.

We hope we have successfully addressed the reviewer’s concerns. Please let me know if you have further questions or comments.

Sincerely,
Rupak Shivakoti, PhD

January 29, 2026

We thank you and the reviewers for the constructive and helpful comments on our manuscript titled “Impact of maternal HIV infection on the gut microbiome and metabolome of mothers and infants: The PRACHITi Cohort in Pune, India.” We have revised our manuscript addressing the additional minor comments from the reviewers as shown in detail below with a point-by-point response. Please note that the number lines refer to the highlighted version of the manuscript.

Reviewer Comments:

Reviewer #2:

1. Is the full cohort made of 245 or 242? The number in the abstract is different from the one in text.

Response: Thank you for asking this question. After checking carefully, our full cohort is actually made of 244 participants (we removed a duplicate third trimester visit), with 242 of them having samples available during pregnancy (with 2 additional women having samples during postpartum but not pregnancy). We have updated lines 131-133 to clarify this further:

“242 women out of 244 women in the full cohort had available samples at the second and/or third trimesters.”

2. The manuscript has several sections saying “data/results not shown”. Could this maybe be shown as supplementary or removed if not important?

Response: This was also an editorial comment in the checklist. We have updated these sections with results (e.g. p-values) where necessary and removed any sections that are not as relevant to include.

3. Is there a reason why in the LMM models, you use different methods for p value adjustment? In others you use Bonferroni and BH in others.

Response: As we state in our paper, the BH adjusted p-values is the focus of our interpretation. However, we also wanted to present the point estimates and 95% confidence intervals, which is useful for visualization purposes. It is not straightforward to adjust the confidence interval when using BH. Thus, following recommendations from this paper based on European Medical Agency guidelines (<https://link.springer.com/article/10.1186/s12874-019-0754-4>), we use point estimates and Bonferroni correction-based confidence intervals only for visualization purposes. Our text in the methods is referring to this in lines 553-555:

“The p-values for taxa were adjusted for multiple comparisons using either the BH method to calculate adjusted p-values (which is the focus of our interpretations) and

Bonferroni method to calculate 95% confidence intervals (which is being used primarily for visualization purposes).”

4. You say in line 314 that WHIV had a lower alpha diversity compared to controls, but the p value is 0.12. You mention that Chao1 and Fisher was significant, but that data is not shown.

Response: Thank you for this comment. WHIV has lower alpha diversity based on the negative coefficient, but we added in parentheses that these results are not significant. We also now present the significant results for Chao1 and Fisher as shown in lines 143-147: “Similar to the results from the sub-study with 71 pregnant women, in a LMM with repeated α -diversity, WHIV had lower α -diversity (although not significant) with HIV status having a coefficient of -0.059 (p=0.12) for Shannon index (Supplementary Table 6), with similar and significant results for Chao1 (p=0.005) and Fisher (p=0.002), adjusting for covariates (same as those for PERMANOVA).”

5. Is the heatmap in Supp Fig 2 complete? You mention 17 taxa but only 3 are shown on rows on the heatmap. The metabolites on columns are also absent and the grouping information on the heatmap is missing.

Response: We have added in our legends for Supplementary Figure 2 the following line to clarify why we only mention a few bacteria and metabolites:

“Due to the large number of metabolites and microbes, we only label select bacteria and metabolites with high positive or negative correlations. The specific correlations with an absolute correlation $\geq \pm 0.3$ between microbes and metabolites are shown in Supplementary Table 9b.”

6. Line 625–628, the discussion of the two taxa here that were differentially abundant in the second and third trimester appear misplaced. This section is for differentially abundant taxa in the postpartum period.

Response: We purposely mentioned these two taxa in the postpartum section because they were the top two taxa in the postpartum results with a significant raw p-value but not adjusted p-value. We thought it was interesting that they were top taxa in the combined second and third trimester results along with the postpartum results. We rephrased the sentence to be clearer for the audience to understand:

Lines 337-340: *“Although not statistically significant, postpartum WHIV had a higher abundance of Megamonas and Lachnoclostridium, bacteria that were significantly higher in WHIV during pregnancy.”*

7. Include references in the discussion, line 723

Response: Thank you for pointing this out. We have now updated our bibliography to reflect only the references we cited in our manuscript.

8. Since you mention the differentially present/absent taxa in the results (and these are different from differentially abundant taxa), could you please discuss in a bit more detail what the implications of such taxa might be for WHIV or their CHEU?

Response: Thank you for this comment and opportunity to clarify further. We have added the following text in lines 557-560 to describe that they are both used together for interpretation:

“Of note, differentially abundant taxa and differentially present/absent taxa refer to distinct analytical approaches used in ANCOM-BC to deal with each type of taxa. However, both are considered together for interpretation of differences between study groups.”

Review of “Impact of maternal HIV infection on gut microbiota and metabolome of maternal-infant populations”

Using a cohort of young low-income pregnant women in Pune, India, the authors investigated how maternal HIV status is associated with gut microbiota at the second and third trimesters of pregnancy and postpartum up to 1 year. They also investigated the maternal plasma metabolomics during the third trimester of pregnancy to understand the differences in the metabolic pathways and functions between women with and without HIV. For a subset of participants, temporal changes in the microbiome and plasma metabolome by HIV status were also investigate. The authors also compared the gut microbiome of infants at 6 months of age by maternal HIV status. Although, this is an interesting study that fills an important gap in the literature regarding changes in the gut microbial ecology during pregnancy, I have some comments as follows.

- 1. Models:** The authors interpreted their data using three different statistical models. Model 0 is the crude model unadjusted for any covariates, Model 1 adjusted for age, education, and anthropometrics, and Model 2 further adjusted Model 1 for smoking status, gestational diabetes (GDM) and anemia. While, I can understand Model 1, but to me Model 2 does not seem appropriate. Please note that HIV condition can disrupt the gut microbiome which in turn can lead to anemia or HIV may have a direct effect leading to anemia. Thus, anemia is potentially a collider and not a confounder. Anemia does not affect HIV status and not sure if it causes changes in the microbial ecology. It will be helpful, if the authors draw a DAG to explain why they should adjust for anemia. For the same reasons, I am not sure about GDM either. Furthermore, the two groups don't seem to be significantly different (Table 1). Since smoking is not very common in this population (Table 1) and the two groups don't appear to be significantly different, I think it is better not to adjust for it in the model. It is sufficient to present results from Model 1 only.
- 2. Sexual activity and substance use.** Major contributors to HIV infection are sexual activity and substance use. It would be helpful if the authors provided information on number of sexual partners the women had and whether any of the partners are known to be people with HIV. If such data are available, then summarize these data in Table 1. Also, cross-tabulate the HIV status with number of sexual partners. This can be relegated to supplementary text. Similarly, it would be useful to provide information on substance use and needle sharing.
- 3. Spaghetti plots:** While the spaghetti plots tracking the alpha diversity over the gestational age between second and third trimesters is interesting to visualize, I find them to be difficult to interpret. Secondly, in a normal pregnancy (without HIV infection or any other health condition), the alpha diversity as well as the microbial

compositions are expected to change during the course of pregnancy (Kore et al., Cell, 2012). The alpha diversity is expected to decrease with gestational age in a normal pregnancy (Mesa et al., Nutrients 2020). In view of this, it would be interesting to test if this hypothesis is true by HIV status. The authors could use the software package CLME (Constrained Linear Mixed Effects) to test whether the mean alpha diversity is decreasing within each HIV group. The CLME software is designed for testing such hypotheses.

4. **Figure 2 (and Supplementary tables 2 and 3):** As noted above in my point #2, the gut microbiota are continuously evolving during the course of pregnancy. In view of that, it does not seem appropriate to combine data from trimesters 2 and 3 and compare the HIV groups as the primary analysis (Figure 2) and treat differential abundance analysis of individual trimesters as secondary analysis (Supplementary tables 2 and 3). Also, it appears some taxa are reported at the family level and others at genus level. Is there a reason to do so? Also, I was surprised to see very few taxa to be differentially abundant, whereas a large number of taxa are differentially abundant at postpartum.
5. **Bacteroides and Prevotella:** Many Bacteroides species are commensal gut bacteria and some of them play an important role in the production of short chain fatty acids. On the contrary Prevotella species are known to be proinflammatory and reported in the literature to have increased abundance in people with HIV. Hence, I was surprised that these taxa were not differentially abundant in this pregnancy cohort. It will be interesting to see the log ratio of Bacteroides to Prevotella is significantly different by HIV status in this population.
6. **Figure 4:** These are interesting postpartum results. I was surprised to see Akkermansia to be the most differentially abundant taxa with higher abundance in women with HIV. It is well-known that Akkermansia Muciniphila plays an important role in the gut health by contributing to the production of short chain fatty acids. I would have expected it to have higher abundance in women without HIV. It will be useful to perform CLME analysis, by HIV status, to see if the alpha diversity decreased from 2nd trimester to third trimester and then increased 6 months postpartum. Thus, a “U” shaped response in mean alpha diversity.
7. **Metabolomics:** Although it appears the authors obtained data from both positive and negative ion channels, it is not clear if they analyzed them separately (which is the preferred approach) or did they combine them into one? Also, in the methods section, they indicate that they adjusted for the “same covariates as in the microbiota analysis”. However, it is not clear to me if the results presented in

Supplementary Table 4 are based on Model 1 or Model 2? I suggest presenting results only from Model 1.

Other comments:

- **Table 1:** shows the distributions at 3rd trimester. Did the HIV status of any participant change from 2nd to 3rd trimester of pregnancy. This might be important to consider because the authors combined 2nd and 3rd trimesters together. Is there a similar table available for 2nd trimester? What about infants?
- **Breastfeeding:** Given that this is a low-income population, due to affordability, I expected almost all babies to breastfed, but it appears the opposite is the case (exclusively breastfed = 60 and not exclusively breastfed = 116).
- **Measures of alpha diversity:** While it is impressive to consider so many different measures of alpha diversity, I think it is sufficient to consider just one measure of their choice and mention that other measures also considered with similar results (data not shown).
- **Non-pregnant women and third trimester:** I am not sure why the authors compared the non-pregnant women with third trimester women. If anything, I would have like to see the results by comparing post-partum data with non-pregnant women data by HIV status. This would tell us whether the postpartum gut ecology is closer to that of non-pregnant women, in the respective HIV groups.